# Structural insights into the allosteric inhibition of P2X4 receptors

Cheng Shen[1,5], Yuqing Zhang[2,5], Wenwen Cui[2,5], Yimeng Zhao[1,3], Danqi Sheng[1], Xinyu Teng[1], Miaoqing Shao[2], Muneyoshi Ichikawa [4], Jin Wang [2] ✉ & Motoyuki Hattori [1] ✉

P2X receptors are ATP-activated cation channels, and the P2X4 subtype plays important roles in the immune system and the central nervous system, particularly in neuropathic pain. Therefore, P2X4 receptors are of increasing interest as potential drug targets. Here, we report the cryo-EM structures of the zebrafish P2X4 receptor in complex with two P2X4 subtype-specific antagonists, BX430 and BAY-1797. Both antagonists bind to the same allosteric site located at the subunit interface at the top of the extracellular domain. Structure-based mutational analysis by electrophysiology identified the important residues for the allosteric inhibition of both zebrafish and human P2X4 receptors. Structural comparison revealed the ligand-dependent structural rearrangement of the binding pocket to stabilize the binding of allosteric modulators, which in turn would prevent the structural changes of the extracellular domain associated with channel activation. Furthermore, comparison with the previously reported P2X structures of other subtypes provided mechanistic insights into subtype-specific allosteric inhibition.

ATP not only serves as the primary source of energy in cells but also acts as an extracellular signaling molecule[1]. P2X receptors are ATP-activated, nonselective cation channels with diverse biological functions[2-4]. The P2X receptor family consists of seven subtypes, designated P2X1-P2X7, that form homo- or heterotrimers[5,6]. While the transmembrane (TM) domain of P2X receptors is responsible for ion permeation, the extracellular domain contains an agonist binding site as well as binding sites for regulatory factors, including divalent cations and chemical compounds[7]. With ubiquitous expression patterns, P2X receptors regulate many essential cellular functions, including synaptic transmission, pain perception, inflammatory response, and smooth muscle contraction[8-10].

Among the seven P2X subtypes, the P2X4 receptor subtype is highly $Ca^{2+}$ permeable and is expressed on a variety of cells, including central neurons and immune cells[11,12]. In particular, P2X4 receptors are

expressed in macrophages, microglia and monocytes, where they participate in the release of proinflammatory cytokines; thus, P2X4 receptors are a promising target for the treatment of neuropathic pain[13-15].

As increasing evidence highlights the role of P2X4 receptors in physiological and pathological processes, several subtype-specific antagonists have been identified[16], including BX430[17] and BAY-1797[18], and some have entered clinical trials for the treatment of neuropathic pain[8].

BX430, or 1-(2,6-dibromo-4-isopropylphenyl)-3-(3-pyridyl)urea, is a phenylurea antagonist. BX430 inhibits human P2X4 currents with an $IC_{50}$ value of 0.54 μM and has no significant effect on other subtypes[17]. In addition, BX430 inhibits both human and zebrafish P2X4 but not rat and mouse P2X4[19]. BAY-1797, or N-[4-(3-chlorophenoxy)-3-sulfamoyl-phenyl]-2-phenylacetamide, is a sulfonamide antagonist of P2X4 with

[1]State Key Laboratory of Genetic Engineering, Shanghai Key Laboratory of Bioactive Small Molecules, Collaborative Innovation Center of Genetics and Development, Department of Physiology and Neurobiology, School of Life Sciences, Fudan University, Shanghai 200438, China. [2]School of Basic Medicine and Clinical Pharmacy, China Pharmaceutical University, Nanjing 211198, China. [3]Human Phenome Institute, Fudan University, Shanghai 201203, China. [4]State Key Laboratory of Genetic Engineering, Department of Biochemistry and Biophysics, School of Life Sciences, Fudan University, Shanghai 200438, China. [5]These authors contributed equally: Cheng Shen, Yuqing Zhang, Wenwen Cui. ✉e-mail: wangjin@cpu.edu.cn; hattorim@fudan.edu.cn

an IC$_{50}$ of 211 nM and high subtype specificity[18]. It shows anti-inflammatory and antinociceptive effects in a mouse model[18].

In efforts to understand the molecular mechanism of P2X receptors, various structures of P2X receptors of different subtypes (P2X3, P2X4 and P2X7) in different functional states (apo-, agonist- and antagonist-bound states) have been reported[20–29]. For the P2X4 receptor, both apo- and ATP-bound structures of the zebrafish P2X4 receptor have been reported[20,21], revealing the channel activation mechanism. In addition, structure determination of P2X3 and P2X7 receptors in complex with antagonists, including negative allosteric modulators, provided the structural basis for the inhibition mechanism of these receptors[23,24,26,29]. However, despite the physiological and pharmacological importance of P2X4 receptors, the detailed mechanism of P2X4 inhibition by antagonists, such as the mechanism of P2X4 subtype-specific allosteric inhibition, remains unclear due to the lack of structures of the P2X4 receptor in complex with its selective antagonists.

Here, we report cryo-electron microscopy (cryo-EM) structures of the zebrafish P2X4 receptor in complex with two different P2X4-selective antagonists, BX430 and BAY-1797. By combining these structures with structure-based patch-clamp analysis, we characterized an allosteric site at the subunit interface in the extracellular domain of the P2X4 receptor. Structural comparison with the previously determined apo structure and molecular dynamics (MD) simulation revealed structural rearrangement at the binding pocket of the allosteric site to stabilize allosteric modulator binding, which is associated with channel inactivation. Further comparison with previously determined P2X structures of other subtypes provided mechanistic insights into subtype-specific inhibition. These results will facilitate the structure-guided design and optimization of allosteric modulators for the P2X4 receptor.

## Results

### Functional characterization and structure determination

We first evaluated the effects of BX430 and BAY-1797 by whole-cell patch-clamp recording of the zebrafish P2X4 (zfP2X4) receptor expressed in HEK293 cells (Supplementary Fig. 1). BX430 inhibited ATP-dependent currents of both zebrafish P2X4 and human P2X4 (hP2X4) with comparable IC$_{50}$ values at sub-micromolar levels (Supplementary Fig. 1A, B), and the results were consistent with the previously reported IC$_{50}$ value of 0.54 μM for hP2X4[17]. While BAY-1797 has been shown to inhibit the hP2X4 receptor[18], it was not clear whether BAY-1797 could act at the zfP2X4 receptor. In our experiments, BAY-1797 inhibited ATP-dependent currents of both zfP2X4 and hP2X4 with comparable IC$_{50}$ values of 0.14 μM and 0.24 μM, respectively (Supplementary Fig. 1C, D).

Next, we used single-particle cryo-EM to determine the structures of the zebrafish P2X4 receptor in complex with BX430 and BAY-1797. The purified zfP2X4 protein was reconstituted into PMAL-C8 amphipol and incubated with 50 μM BX430 or 125 μM BAY-1797 for single-particle cryo-EM analysis. After cryo-EM data acquisition and image processing, the final EM map resolutions were 3.2 Å for the BX430-bound structure and 3.4 Å for the BAY-1797-bound structure (Supplementary Figs. 2–5). Notably, we also processed our cryo-EM data with C1 symmetry (Supplementary Fig. 6) and found that the structures still showed symmetric trimers but at lower resolutions (3.9 Å resolution for the BX430-bound structure and 4.0 Å resolution for the BAY-1797-bound structure).

### Allosteric binding site

The overall structures of the zebrafish P2X4 receptor in complex with BX430 and BAY-1797 are very similar (Supplementary Fig. 7A) and show a chalice-like trimeric architecture with each subunit containing a relatively large extracellular domain and two TM helices (Fig. 1), resembling the shape of a dolphin (Supplementary Fig. 8A), as in the

previously reported P2X receptor structures[30]. Consistent with the function of BX430 and BAY-1797 as antagonists, the TM helices of both structures adopt a closed conformation of the channel (Supplementary Fig. 7B, C).

In the extracellular domain of the two structures, we observed residual EM densities that matched the shapes of BX430 and BAY-1797 at the same location of the intersubunit interface of the trimer (Fig. 1). In the dolphin model, this binding site is located in the upper body domain and is distant from the ATP binding site, defining the binding site as an allosteric site (Supplementary Fig. 8A, B). Consistently, upon the binding of BX430, there were no structural changes that might inhibit the binding of ATP (Supplementary Fig. 7D).

To verify the P2X4 structures in complex with BX430 and BAY-1797, we performed MD simulations. During the MD simulations, both overall structures were mostly stable (Supplementary Fig. 9A, 9C), and both BX430 and BAY-1797 were stably bound to the allosteric site (Supplementary Fig. 10A, B). In addition, we generated a model of human P2X4 in complex with BX430 based on our cryo-EM structure of zebrafish P2X4 in complex with BX430 and performed MD simulation. Similar to the results of MD simulation of the zebrafish P2X4 structures, the overall structure of human P2X4 was mostly stable (Supplementary Fig. 9E), and the BX430 molecules were stably bound to the allosteric site (Supplementary Fig. 10C).

In the BX430-bound structure, the dibromo-isopropylphenyl group of BX430 faces the outer side of the receptor, while the pyridine group faces the inner side (Fig. 2). The binding site consists mainly of hydrophobic residues, including Trp87, Ile94, Met108, Ile110, Phe299, and Ile315 from one subunit and Tyr302 from the neighboring subunit (Fig. 2b, c and Supplementary Fig. 11A). In addition, the main chain carbonyl group of Asp91 forms hydrogen bonds with the amines in the urea group of BX430 (Fig. 2b, c and Supplementary Fig. 11A). Interestingly, the side chain of Lys301 forms a cation-pi interaction with the aromatic ring of Tyr302 from the neighboring subunit, allowing hydrophobic interactions between the side chain of Lys301 and the 4-isopropylphenyl ring of BX430 (Fig. 2b, c and Supplementary Fig. 11A).

Furthermore, there are intersubunit salt bridges between the side chains of Arg85 of one subunit and Glu310 of the neighboring subunit, and hydrogen bonds between the side chain of Tyr302 of one subunit and the main-chain carbonyl group of Ala90 and the side chain of Asp91 of the neighboring subunit (Fig. 2b and Supplementary Fig. 11A). These interactions also seemingly stabilize the formation of the allosteric binding pocket.

In the BAY-1797-bound structure, the phenylacetamide moiety faces the outer side of the receptor, while the chlorophenoxy moiety faces the inner side (Fig. 3). The BAY-1797 binding site also consists primarily of hydrophobic residues (Fig. 3b, d and Supplementary Fig. 11B), which are the same as those of the BX430 binding site (Fig. 2b). Furthermore, both Asp91 and Lys301 are similarly involved in BAY-1797 binding (Fig. 3b and Supplementary Fig. 11B), despite the structural differences between BAY-1797 and BX430. The main chain carbonyl group of Asp91 forms a hydrogen bond with the amine in the phenylacetamide moiety of BAY-1797, and the hydrophobic part of the Lys301 side chain interacts with the phenyl group of the phenylacetamide moiety by forming a cation-pi interaction between the side chain of Lys301 and the aromatic ring of Tyr302 (Fig. 3b and Supplementary Fig. 11B).

While most of the residues involved in BAY-1797 binding overlap with those involved in BX430 binding (Fig. 2e), there are also specific interactions in the binding of BAY-1797. The amine of the sulfonamide group forms hydrogen bonds with the main chain carbonyl groups of Ala90 and Ile93 (Fig. 3b). Finally, the chlorophenoxy moiety of BAY-1797 protrudes into the center of the receptor and forms hydrophobic contacts with the corresponding portion of the other two BAY-1797 molecules in the trimer (Fig. 3c). From another perspective, while the

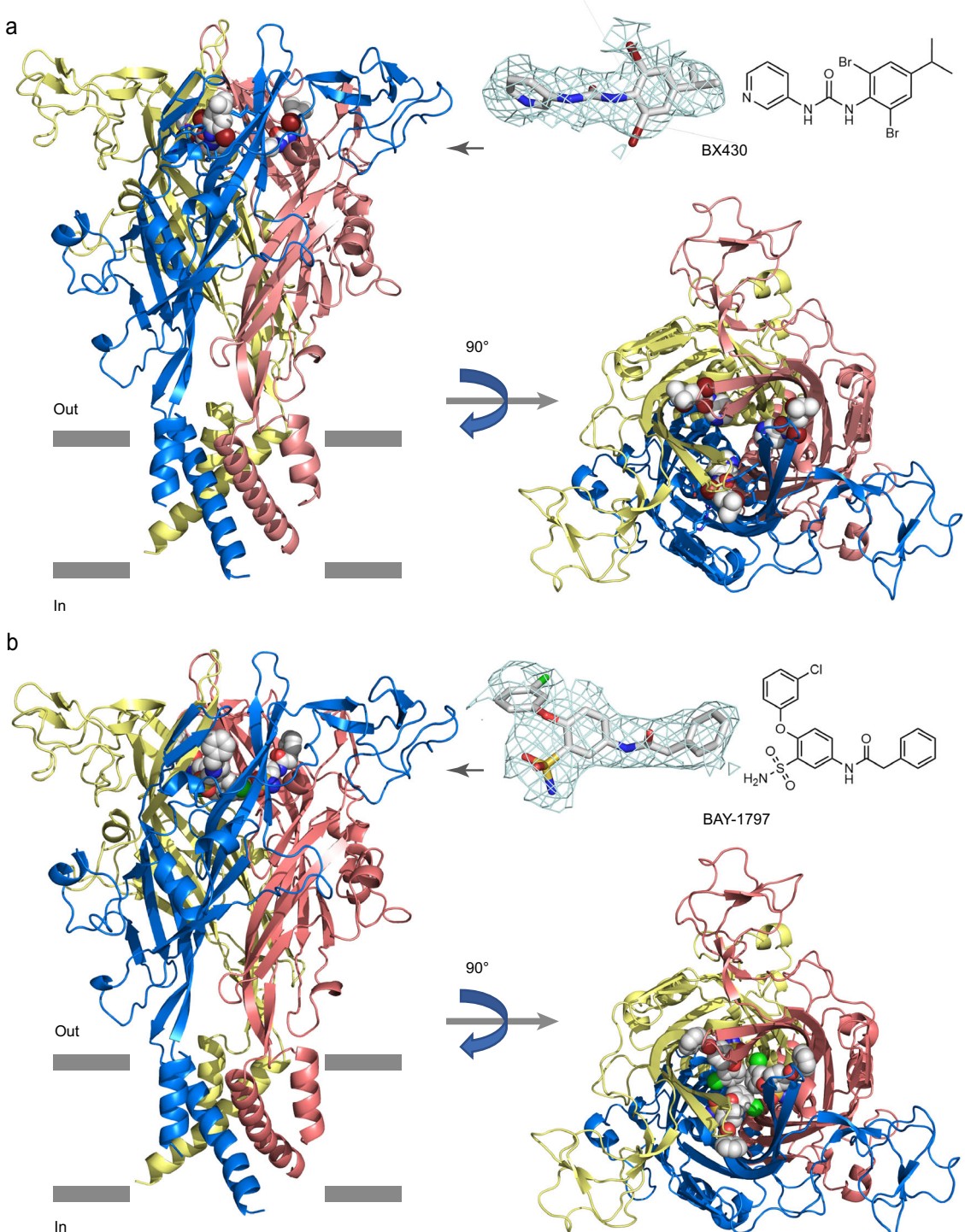

**Fig. 1 | Overall structures.** The cryo-EM structures of the zebrafish P2X4 receptor in complex with BX430 (**a**) and BAY-1797 (**b**), viewed parallel to the membrane (left panel) and perpendicular to the membrane from the extracellular side (right panel). BX430 and BAY-1797 molecules bound to the P2X4 receptor are shown in sphere representations. 2D structures of BX430 and BAY-1797 are also shown. The protomers of the trimer are colored blue, yellow, and red. The EM density maps for BX430 and BAY-1797 are also shown and contoured at 3.0 σ.

configurations of BX430 and BAY-1797 in the zfP2X4 structure are very similar (Fig. 2d), BAY-1797 has bulkier functional groups deeper in the receptor that mediate the BAY-1797-specific interactions (Fig. 3b, c).

### Structure-based mutational analysis

To validate the allosteric site in our structures (Figs. 2, 3), we designed a series of zebrafish P2X4 receptor mutants (R85A, W87A, I94A, I94W, K301V, K301R, Y302A, and E310K) together with the human P2X4

receptor mutants for patch clamp recordings in HEK293 cells (Figs. 2e, 4).

We designed mutations to residues with smaller side chains (mostly to Ala) to reduce the side chain-mediated interactions between the receptor and BX430, which would weaken the binding affinity between them. We designed other types of mutations, such as the introduction of a bulkier side chain (e.g., to Trp) or changing the charge of the side chains (e.g., from Glu to Lys), to introduce repulsion

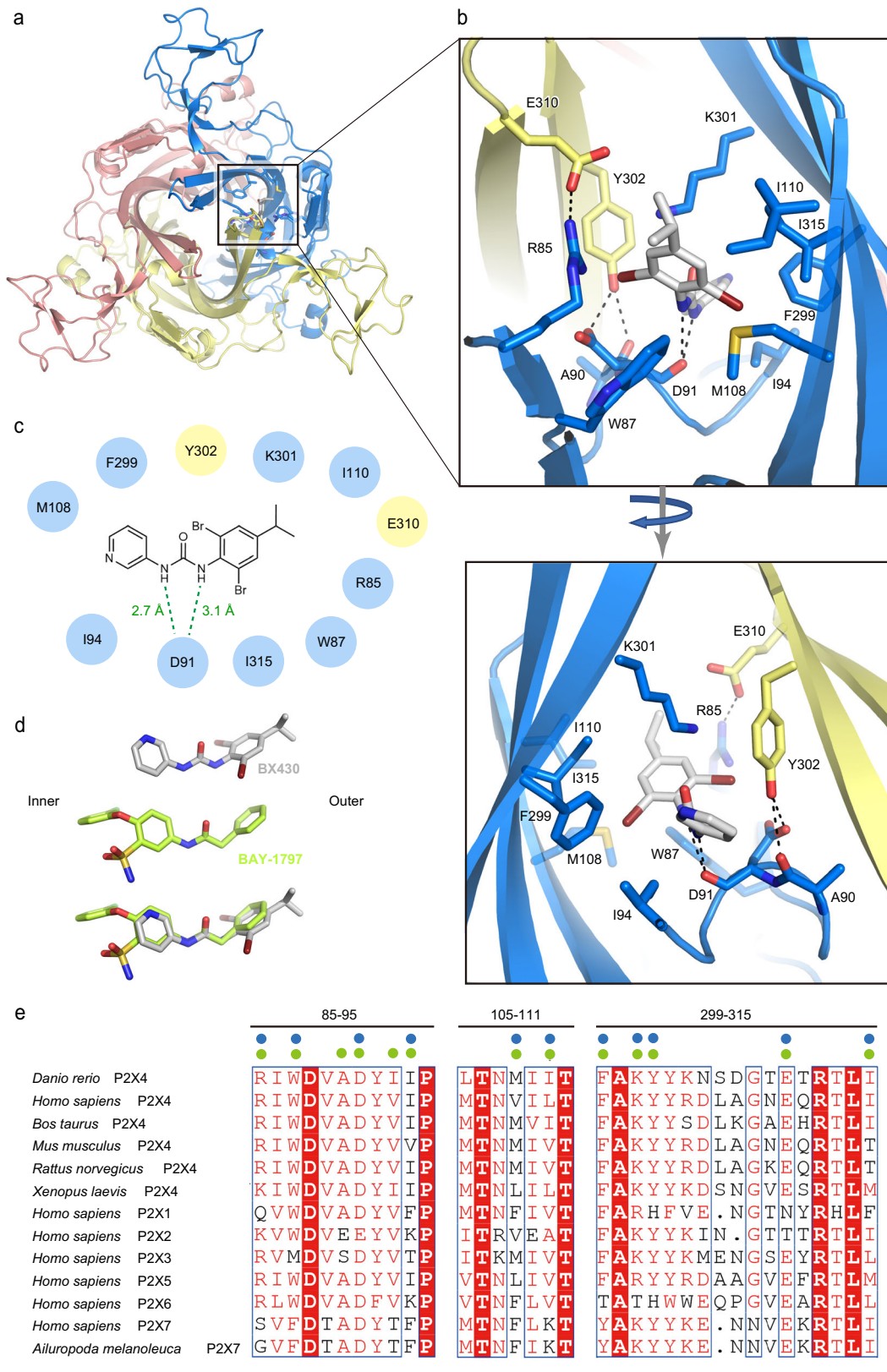

between the receptor and BX430, which would also weaken the binding affinity between them. In addition, some human P2X4 mutants were designed to introduce the types of residues found in other P2X subtypes.

Notably, the mutations at Trp87, Ile94, and Lys301 of zfP2X4 target the residues involved in direct interactions with allosteric modulators, whereas the mutations at Arg85, Tyr302, and Glu310

destabilize the formation of the binding pocket (Figs. 2, 3). Consistent with our structures, all mutants of zfP2X4 showed significantly decreased sensitivity to BX430 compared to the wild-type construct (Fig. 4a, c).

Next, to verify whether this allosteric site is functionally conserved in the human P2X4 receptor, we also performed patch-clamp recording of hP2X4 and its mutants expressed in HEK293 cells

**Fig. 2 | BX430 binding pocket. a** Overall structure of the BX430-bound P2X4 receptor viewed from the extracellular side. The BX430 molecule and amino acid residues involved in its binding from one of three equivalent binding pockets are shown in stick representation. **b** Close-up views of the BX430 binding pocket. Dotted lines represent hydrogen bonds. **c** Schematic diagram of the interactions between zfP2X4 and BX430. Hydrogen between the amine groups of BX430 and the main-chain carbonyl group of zfP2X4 are shown as dotted lines. **d** Close-up views of BX430 and BAY-1797 molecules in the cryo-EM structure and the super-position of BAY-1797 onto BX430. **e** Amino acid sequence alignment of P2X4 receptors from *Danio rerio* (zebrafish) (AAH66495.1), *Bos taurus* (NP_001029221.1), *Mus musculus* (NP_035156.2), *Rattus norvegicus* (AAA99777.1), and *Xenopus laevis* (NP_001082067.1), P2X receptors from *Homo sapiens* (P2X1: P51575.1, P2X2: Q9UBL9.1, P2X3: P56373.2, P2X4: Q99571.2, P2X5: Q93086.4, P2X6: O15547.2, and P2X7: Q99572.4), and *Ailuropoda melanoleuca* (giant panda) P2X7 receptor (XP_002913164.3). The residues of the regions located at the binding pocket are shown. Blue and green circles indicate the residues involved in the binding of BX430 and BAY-1797, respectively.

(Figs. 2e, 4b, d). The residues involved in the allosteric binding site are highly conserved or at least similarly conserved in both human and zebrafish P2X4 receptors (Fig. 2e), and we generated R82A/G/Q (Arg85 in zfP2X4), W84A (Trp87 in zfP2X4), D88W (Asp91 in zfP2X4), I91A/T (Ile94 in zfP2X4), F296W (Phe299 in zfP2X4), K298V/R (Lys301 in zfP2X4), Y299A (Tyr302 in zfP2X4) and E307N/T/W (Glu310 in zfP2X4).

Except for the D88W and F296W mutants (Fig. 4b, d), most of the mutated residues in hP2X4 correspond to the mutated residues in zfP2X4 (Fig. 4a, c). In the BX430-bound structure, the side chain of Asp91 (Asp88 in hP2X4) forms a hydrogen bond with the side chain of Tyr302, contributing to the formation of the binding pocket (Fig. 2b). The Phe299 residue (Phe296 in hP2X4) mediates the hydrophobic interaction with BX430 (Fig. 2b). In the mutational analysis of hP2X4, except for Glu307 mutants, all mutants showed significantly reduced sensitivity to BX430 (Fig. 4c). Among the Glu307 mutants of hP2X4, while the E307W mutant showed weakly reduced sensitivity to BX430, the E307N and 307T mutants showed little change, possibly because these mutations may not be sufficient to break the hydrogen bond with Arg82 in hP2X4 (Arg85 in zfP2X4).

In addition to our mutational analysis, one of the previous studies showed that Ile312 in human P2X4 (Ile315 in zebrafish P2X4) is also important in BX430 sensitivity to human and zebrafish P2X4 as well as in BX430 insensitivity to rat and mouse P2X4, where the corresponding residue is replaced by threonine[19]. These results are consistent with our cryo-EM structure where Ile315 contacts BX430 via hydrophobic interactions (Fig. 2). In addition, in another previous study[31], the potential cavity at the dorsal fin was predicted as a potential binding site for BX430, and the alanine substitution of Ile209 at the dorsal fin of human P2X4 was shown to decrease the BX430 sensitivity. In our cryo-EM map, we did not find the EM density for BX430 in the dorsal fin region and found that the side chain of Ile212, corresponding to Ile209 in human P2X4, has extensive hydrophobic contacts with the neighboring residues (Supplementary Fig. 11C, D). Therefore, a possible explanation for the reduced BX430 sensitivity of the I209A mutant could be that a mutation at Ile209, which mediates extensive hydrophobic interactions with neighboring residues, could affect the folding of the protein and thus indirectly affect the affinity to BX430 at the distant allosteric site in this study. Nevertheless, we do not completely exclude the possibility that another binding site for BX430 may exist, depending on the experimental conditions.

Taken together, these results validated the allosteric site of our structures in both zebrafish and human P2X4 receptors.

### Structural reorganization of the binding pocket

Structural comparison with the previously reported apo structure revealed that in the apo state, the intersubunit binding pocket is too narrow to accommodate allosteric modulators (Fig. 5a–c). Further close inspection of the allosteric site revealed that in the apo state, the side chain of Lys301 faces the inside of the receptor, which would cause steric hindrance with the pyridine group of BX430 in the BX430-bound structure (Fig. 5d, e). In contrast, in the BX430-bound and BAY-1797-bound structures, the side chain of Lys301 moves away from the inside of the binding pocket to form a cation-pi interaction with the aromatic ring of Tyr302 from the adjacent subunit and to form hydrophobic interactions with either the 4-isopropylphenyl ring of

BX430 or the phenyl ring of the phenylacetamide moiety of BAY-1797 (Fig. 5f, g, Supplementary Movie 1).

In other words, the structural changes in the side chain of Lys301 not only provide enough space at the inner side of the binding pocket to accommodate allosteric modulators (Fig. 5e) but also contribute to the direct contacts between Lys301 and allosteric modulators in the outer region of the binding pocket (Fig. 5f, g). Consistently, the mutations at Lys301 and Tyr302 in zebrafish P2X4 and at the corresponding residues in human P2X4 showed decreased sensitivity to BX430 in patch-clamp recording (Fig. 4). Furthermore, the inter-subunit cation-pi interaction between the side chain of Lys301 and the aromatic ring of Tyr302 was maintained during the MD simulations of the BX430-bound and BAY-1797-bound structures (Fig. 5h, i).

### Expansion of the binding pocket for allosteric inhibition

In addition to the structural change of Lys301, structural comparison with the previously reported apo structure revealed the expansion of the binding pocket (Fig. 6a, Supplementary Movie 1). For instance, in the apo structure, the distance between the Cα atoms of Ile110 (chain A) and Gln116 (chain B), located at the entrance region of the binding pocket, is 17.0 Å, whereas the corresponding distances are 18.3 Å and 18.5 Å for the BX430-bound and BAY-1797-bound structures, respectively, reflecting the expansion of the binding pocket (Fig. 6a). The relatively small structural changes at the entrance of the binding pocket upon binding of compounds suggest that such small changes at the entrance of the binding pocket are sufficient to accommodate the compounds (Fig. 6a).

To test whether this expansion is ligand-binding dependent, we performed MD simulations using the BX430-bound structure with BX430 deleted and the BAY-1797-bound structure with BAY-1797 deleted as starting models. In both runs of the MD simulations embedded in lipids, both overall structures were mostly stable (Supplementary Fig. 9B, D). Consistent with our notion, while the binding pocket remains expanded during the MD simulations of the BX430-bound and BAY-1797-bound structures (Fig. 6b, d), the removal of BX430 and BAY-1797 leads to shorter distances of the Cα atoms of Ile110 and Gln116 of two adjacent subunits of the trimer, indicating a closing motion of the binding pocket in the absence of allosteric modulators (Fig. 6c, e). It should be noted that the binding pocket is more expanded in the presence of BX430 than in the presence of BAY-1797 (Fig. 6b, d), probably simply because the 4-isopropylphenyl ring of BX430, located in the outer region of the binding pocket, is bulkier than the phenyl ring of the phenylacetamide moiety of BAY-1797 (Fig. 5f, g).

Notably, our cryo-EM structure is determined in amphipol, whereas the previously reported apo structure was determined by X-ray crystallography in detergents. Therefore, we cannot exclude the possibility that the conformational differences between the BX430/BAY-1797-bound structures and the previously determined apo structure may be due to differences in experimental conditions. Therefore, we performed MD simulation of the previously reported apo structure of zebrafish P2X4 (PDB ID: 4DW0) by embedding it in lipids. The structure was stable during the MD simulation (Fig. 6f) and adopted the nonexpanded conformation of the binding pocket after the MD simulation (Fig. 6g). Importantly, we observed similar expansion of the binding pocket and associated movement of the head domain in both

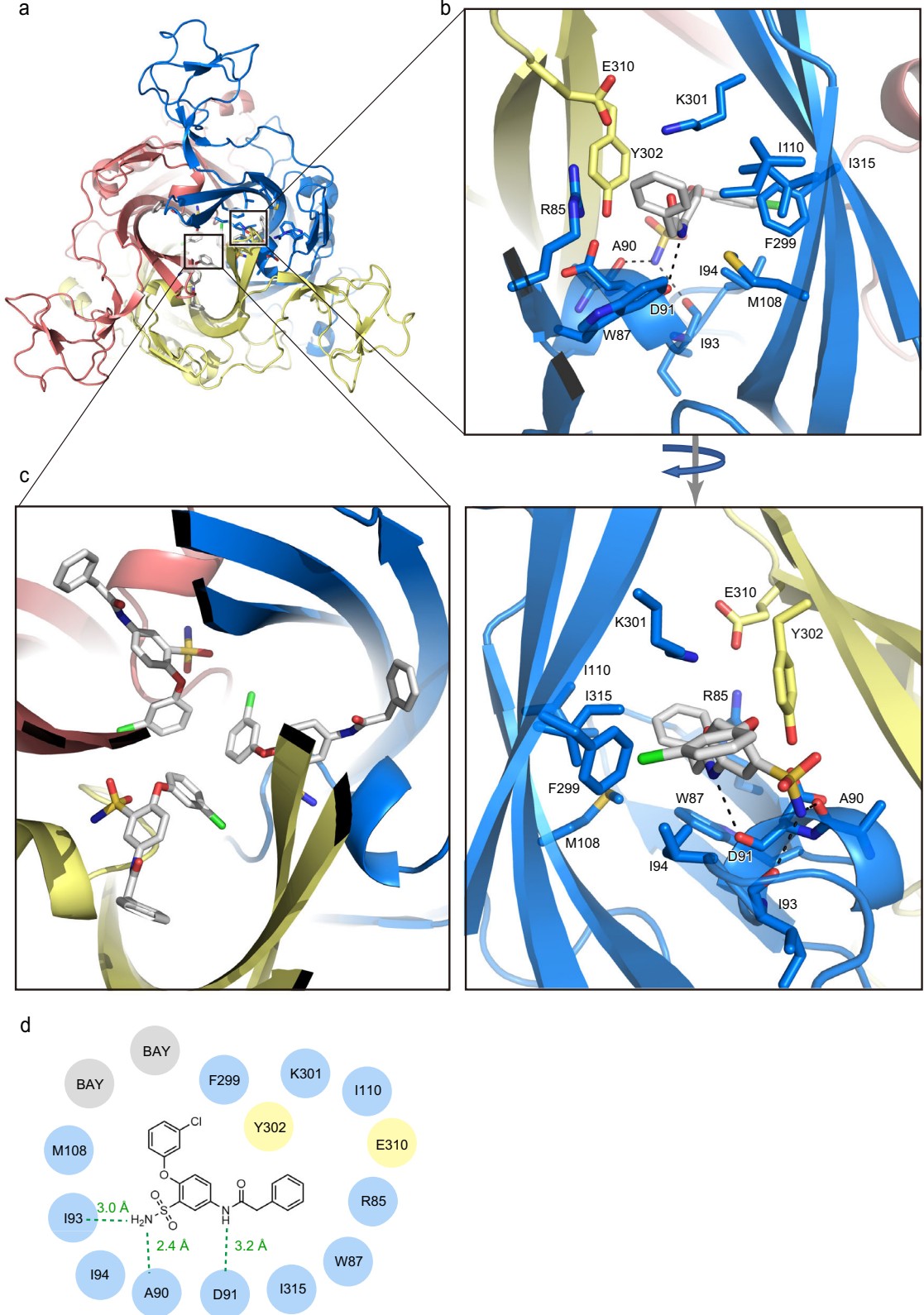

**Fig. 3 | BAY-1797 binding pocket. a** Overall structure of the BAY-1797-bound P2X4 receptor viewed from the extracellular side. The BAY-1797 molecule and amino acid residues involved in its binding in one of three equivalent binding pockets are shown in stick representation. **b, c** Close-up views of the BAY-1797 binding pocket. **b** Dotted lines represent hydrogen bonds. **c** The BAY-1797 molecules from two other equivalent binding pockets are also shown. **d** Schematic diagram of the interactions between zfP2X4 and BAY-1797. Hydrogen between the amine groups of BAY-1797 and the main-chain carbonyl group of zfP2X4 are shown as dotted lines.

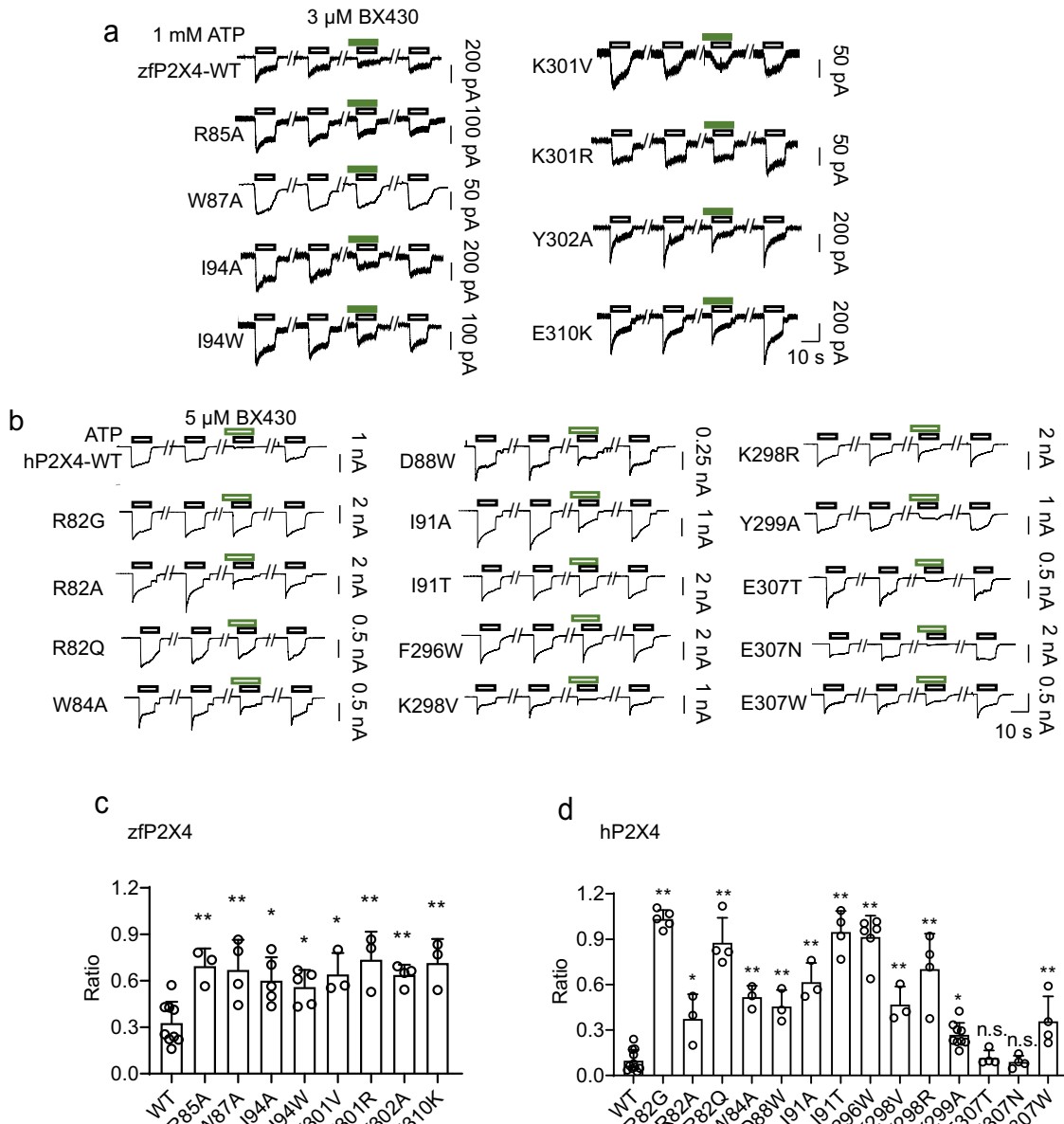

**Fig. 4 | Structure-based mutational analysis. a** Representative current traces of BX430 effects on ATP-evoked currents of zebrafish P2X4 and its mutants. **b** Representative current traces of BX430 effects on ATP-evoked currents of human P2X4 and its mutants. **c** Effects of BX430 (3 μM) on ATP (1000 μM)-evoked currents of zebrafish wild-type P2X4 and its mutants (mean ± SD, $n = 3$ to 9). The value of $n$ is 9, 3, 4, 5, 5, 3, 3, and 4 in ascending order on the x-axis. The p-value is 0.0037, 0.0025, 0.0111, 0.0413, 0.0161, 0.0011, 0.0070, and 0.002, respectively (from left to right). **d** Effects of BX430 (5 μM) on ATP (100 μM)-evoked currents of human P2X4 and its mutants (mean ± SD, $n = 3$ to 11). The value of $n$ is 11, 5, 3, 4, 3, 3, 3, 4, 6, 3, 4, 9,

4, 4, and 4 in ascending order on the x-axis. $n$ is the number of biologically independent cells. The p-value is <0.0001, 0.009, <0.0001, <0.0001, 0.0003, <0.0001, <0.0001, <0.0001, 0.0002, <0.0001, 0.0257, 0.9997, 0.9999, and 0.0052, respectively (from left to right) (one-side one-way ANOVA followed by post hoc test, *$p < 0.05$, **$p < 0.01$ vs. WT). For the additional mutational analysis performed during the revision of the manuscript (R82G, R82Q, I91T, E307N, E307T), 10 μM ATP concentration was tested. The inhibition ratio is defined by normalizing the peak current amplitude from the coapplication of BX430 and ATP to the peak current amplitude from the ATP application prior to the coapplication of BX430 and ATP.

the experimental structures and the MD simulations (Fig. 6h, i). Although there are differences in the experimental conditions between our cryo-EM structures and the previously reported apo structure, MD simulations of the apo crystal structure and the BX430-bound cryo-EM structure were performed in lipid environments, and we still observed the structural differences between them as in the experimental structures. Therefore, it is less likely that these structural differences are due to differences in the solution environment of structure determination (detergent and amphipol) or the methods of structure determination (crystallization and cryo-EM). As the binding pocket in the body domain expands, the head domain adjacent to the body domain also moves away from the center of the receptor (Fig. 7a,

Supplementary Movie 1), which is distinct from the downward movement of the head domain upon ATP-dependent channel activation (Fig. 7b). For instance, the distance between the Cα atoms of Thr149 in the head domain of the apo- and ATP-bound structures is 0.8 Å, whereas the corresponding distances for the BX430-bound and BAY-1797-bound structures are 2.3 Å and 2.1 Å, respectively, reflecting the movement of the head domain. Furthermore, since the upper body domain acts as a fulcrum in the opening motion of the lower body domain during ATP-dependent activation of P2X4 receptors as well as other P2X subtypes[21,28,29] (Fig. 7b), the intersubunit binding of allosteric modulators to the upper body domain would interrupt the structural changes of the body domain (Fig. 7a).

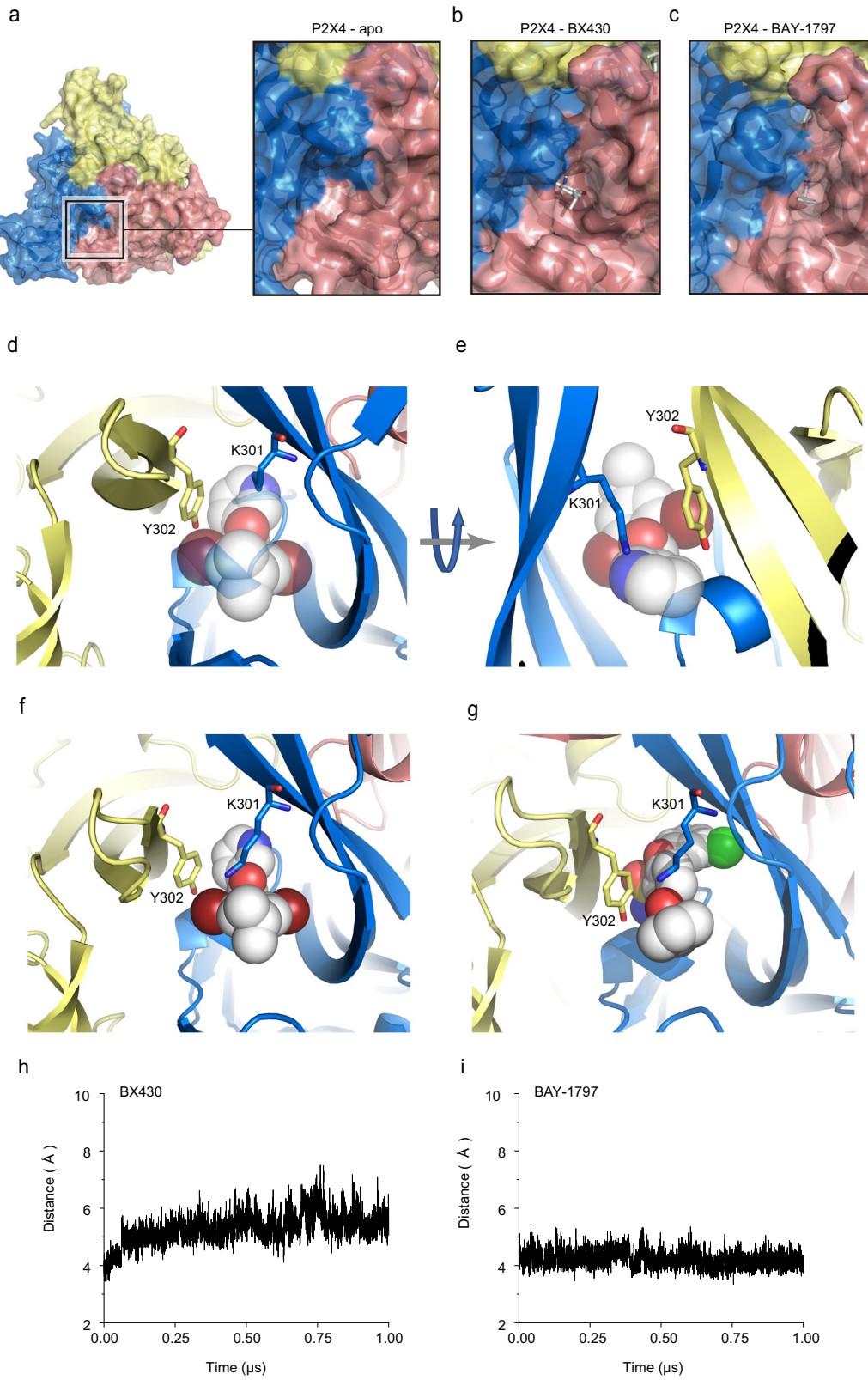

**Fig. 5 | Ligand-dependent structural reorganization of Lys301. a–c** Close-up views of the allosteric site of zfP2X4, viewed from the extracellular side, are shown in surface representations. **a** The zfP2X4 structure in the apo state (PDB ID: 4DW0). **b** The BX430-bound zfP2X4 structure (this study). **c** The BAY-1797-bound zfP2X4 structure (this study). **d–g** Allosteric sites of zfP2X4 in the apo structure (PDB ID: 4DW0) (**d, e**), in the BX430-bound structure (**f**), and in the BAY-1797 bound structure (**g**). The BX430 (**f**) and BAY-1797 (**g**) molecules are shown in sphere representation. The BX430 molecule from the BX430-bound structure superposed onto the apo structure is shown in half-transparent sphere representation (**d, e**). **h, i** MD simulations of the BX430-bound structure (**h**) and the BAY-1797-bound structure (**i**). Distance plots between the NZ atom of Lys301 and the center of the aromatic ring of Tyr302. The averaged distance values from the three subunits of the trimer are shown. Two more additional repeats of the MD simulations were shown in Supplementary Fig. 13M–P. In total, MD simulations were performed three times.

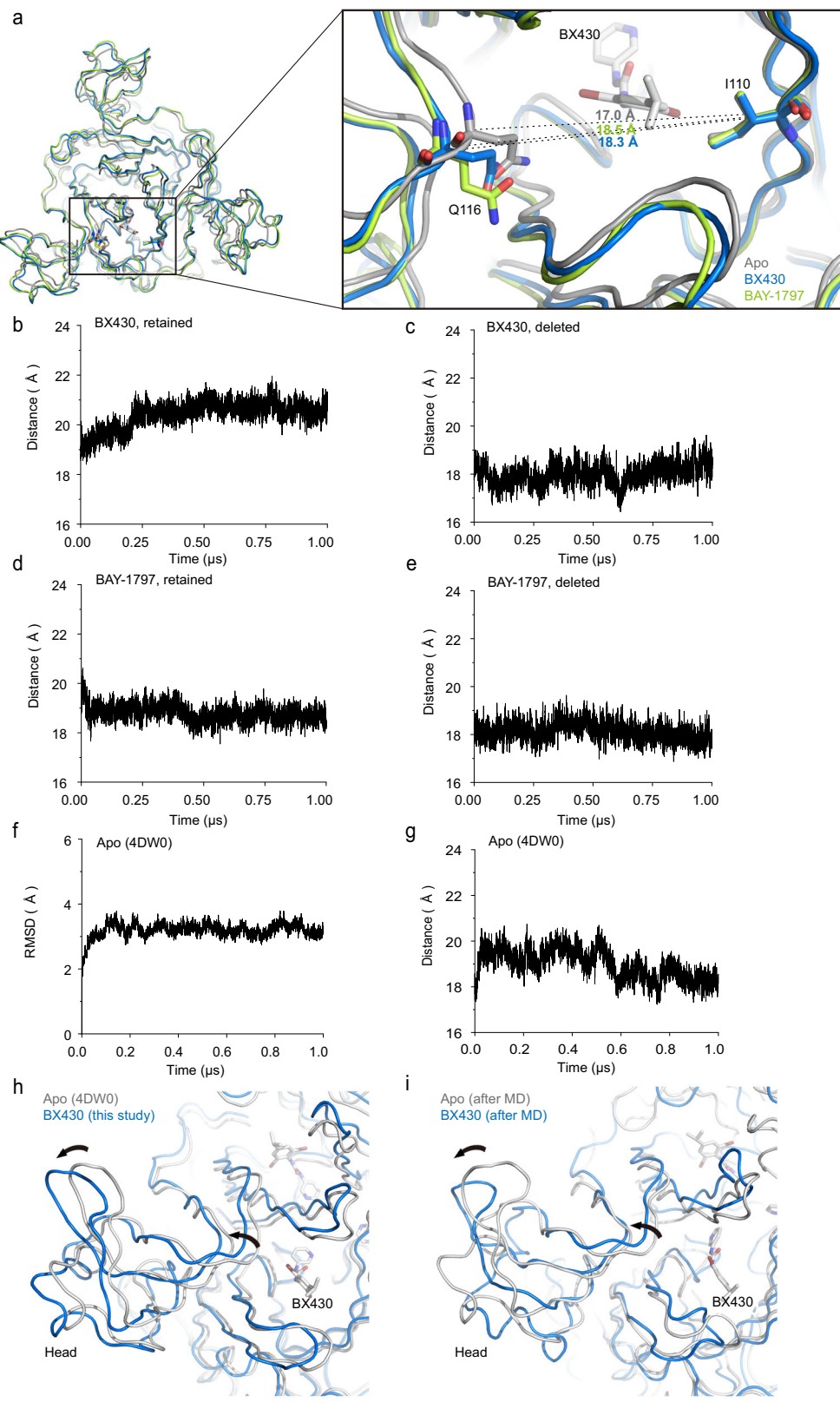

In summary, the allosteric modulator-dependent expansion of the binding pocket of the P2X4 receptor induces structural changes in the head domain and provides conformational stabilization of the upper body domain through the accommodation of modulators, which in turn prevents the structural changes associated with channel activation (Fig. 7c).

## Discussion

In this work, we determined the cryo-EM structures of the zebrafish P2X4 receptor in complex with two different negative allosteric modulators, BX430 and BAY-1797 (Fig. 1), and performed structure-based mutational analysis of the allosteric site by patch-clamp recording of both zebrafish and human P2X4 receptors (Figs. 2–4).

**Fig. 6 | Ligand-dependent expansion of the binding pocket. a** Close-up view of the allosteric site of the zfP2X4 receptor. The BX430-bound (blue) and BAY-1797-bound (green) structures are superposed onto the apo structure (PDB ID: 4DW0, gray). Dotted lines indicate the distance (Å) between the Cα atoms of Ile110 and Gln116 of two adjacent subunits. **b**−**e** MD simulations using the BX430-bound structure with BX430 retained (**b**) or deleted (**c**) and the BAY-1797-bound structure with BAY-1797 retained (**d**) or deleted (**e**) as starting models. The distance plots of Cα atoms between Ile110 and Gln116 of two adjacent subunits are shown. The averaged distances from the three subunits of the trimer are shown. The distance values averaged over the entire 1-μs runs are 20.4 Å (**b**), 18.0 Å (**c**), 18.8 Å (**d**), and 18.1 Å (**e**). Two more additional repeats of the MD simulations were shown in Supplementary Fig. 13Q–X. The distance values averaged from three independent runs are 20.3 ± 0.4 Å for (**b**), 18.1 ± 0.1 Å for (**c**), 19.3 ± 0.6 Å for (**d**), and 18.4 ± 0.6 Å for (**e**) (two-sided unpaired *t*-test, *p* = 0.0005 between (**b**) and (**c**), *p* = 0.1702 between (**d**) and (**e**)). **f, g** MD simulations of the previously published apo structure of zfP2X4 (PDB ID: 4DW0). The plots of the root mean square deviations (RMSD) for Cα atoms (**f**) and distance of Cα atoms between Ile110 and Gln116 of two adjacent subunits (**g**) are shown. Two more additional repeats of the MD simulations were shown in Supplementary Fig. 15. In total, MD simulations were performed three times. **h** Superimposition of the BX430-bound zfP2X4 structure (this study, blue) onto the apo zfP2X4 structure (PDB ID: 4DW0, gray). **i** Superimposition of the BX430-bound zfP2X4 structure (after the MD simulation, blue) onto the apo zfP2X4 structure (after the MD simulation, gray). Arrows indicate structural changes at the binding pocket and the head domain.

Comparison with the previously reported apo structure revealed the structural rearrangement of Lys301 (Fig. 5), the mutation of which decreases sensitivity to BX430 (Fig. 4). Further structural comparison and MD simulations revealed the ligand-dependent expansion of the binding pocket (Fig. 6), which would be associated with the allosteric inhibition of channel activation (Fig. 7).

Several previous studies have characterized BX430 by using electrophysiological analysis and in silico docking simulations[19,31,32]. Notably, two independent groups have used electrophysiology to identify Ile312 in human P2X4 (Ile315 in zebrafish P2X4) as a key residue in the formation of a binding pocket for allosteric inhibition, but their BX430 docking simulation results differ[19,32]. In an earlier study[19], the dibromo-isopropylphenyl group of BX430 was oriented toward the inner side of the receptor, which contrasts with our cryo-EM structure (Fig. 2). In a more recent report of induced-fit docking[32], the dibromo-isopropylphenyl group of BX430 was oriented toward the outer side of the receptor. However, in the induced-fit docking, Lys298 in human P2X4 (Lys301 in zebrafish P2X4) did not interact with either BX430 or Tyr299 in human P2X4 (Tyr302 in zebrafish P2X4) but rather pointed away from them, which does not agree with our cryo-EM structure and mutational analysis (Figs. 4, 5). In our cryo-EM structure, Lys301 forms a cation-pi interaction with the aromatic ring of Tyr302 and a hydrophobic interaction with the 4-isopropylphenyl ring of BX430 (Fig. 5), and the mutations at Lys301 in zebrafish P2X4 and Lys299 in human P2X4 reduced the sensitivity to BX430 (Fig. 4). Furthermore, it was also predicted that ligand binding would disrupt the important interaction networks of Asp85, Ala87, Asp88 and Ala297 of human P2X4[32], which are known to be essential for ATP-dependent activation (Asp88, Ala90, Asp91 and Ala300 in zebrafish P2X4), but this prediction also contrasts with our cryo-EM structure. This discrepancy may also be due to the difference in Lys301, which is adjacent to Asp88, Ala90, Asp91 and Ala300. Another recent report predicted the region in the dorsal fin domain to be a potential BX430 binding site, which is also inconsistent with our cryo-EM structure[31].

Comparison of the allosteric site in our structures with the corresponding region in previously reported P2X structures provided mechanistic insight into the subtype specificity of allosteric modulators for the P2X4 receptor (Fig. 8). First, the comparison of the BX430-bound zfP2X4 structure with the previously reported P2X3 structure showed that while most of the residues at the allosteric site are common (Fig. 2e), Trp87, Ile94 and Ile110 in zfP2X4 are replaced by Met75, Thr82 and Val98 in human P2X3 (Fig. 8a), respectively, which would negatively affect the interactions with BX430. Consistently, the mutations at residues Trp87 and Ile94 in zfP2X4 and the mutations at the corresponding residues of hP2X4 caused lower sensitivity to BX430 (Fig. 4). Notably, the P2X3-like mutant of hP2X4 I91T (Ile94 in zfP2X4) showed significantly lower sensitivity to BX430 (Figs. 4d, 8a).

Next, the comparison of the BX430-bound zfP2X4 structure with the previously reported P2X7 structure showed that Arg85 and Ile110 in zfP2X4 are replaced with the Gly and Lys residues in the panda P2X7 structure, respectively (Fig. 8b), which would cause the loss of

the salt bridge with the glutamate residue as well as steric hindrance with BX430. In our mutational analysis, the mutation at Arg85 in zfP2X4 and the mutations at the corresponding residue of hP2X4 decreased the sensitivity to BX430 (Fig. 4). In particular, the P2X7-like mutant of hP2X4 R82G (Arg85 in zfP2X4) (Figs. 4d, 8b) showed significantly lower sensitivity to BX430.

In addition, since no experimental structures have been reported for the other functionally well-characterized P2X receptors (P2X1 and P2X2), we generated predicted structural models of human P2X1 and P2X2 using AlphaFold and ColabFold and superposed them onto the BX430-bound structure of zfP2X4 (Fig. 8c, d). For human P2X1 receptor, Arg85, Ile94, Lys301, Tyr302, Glu310, and Ile315 at the allosteric site of zfP2X4 correspond to Gln83, Phe92, Arg295, His296, Asn303, and Phe308 in human P2X1, respectively (Fig. 8c) and are shown to be important for BX430 sensitivity either by our mutational analysis (Fig. 4) or by the previous mutational analysis targeting the residue corresponding to Ile315[19,32]. Notably, the P2X1-like mutant of hP2X4 R82Q (Arg85 in zfP2X4) showed significantly lower sensitivity to BX430 (Figs. 4d, 8c). For the human P2X2 receptor, Arg85, Ile94, Ile110 and Glu310 in zfP2X4 are replaced by Lys91, Lys100, Ala116 and Thr313 in human P2X2, respectively (Fig. 8d). Among them, the mutations at Arg85, Ile94 and Glu310 in zfP2X4 and the mutations at Arg82 and Ile91 in hP2X4 resulted in decreased sensitivity to BX430 (Fig. 4), whereas the mutations of hP2X4 at Glu307 (Glu310 in zfP2X4) showed weak or little decrease in sensitivity to BX430 (Fig. 4d).

Finally, to further deepen the mechanistic insights into subtype specificity, we designed a gain-of-function mutant of human P2X3 to confer BX430 sensitivity to human P2X3 by introducing P2X4-like mutations into P2X3 (Fig. 8e, f). Strikingly, while the human P2X3 receptor is known to be insensitive to BX430, the P2X4-like mutant of human P2X3 (M75W/T82I/V98I) conferred sensitivity to BX430 (Fig. 8e, f).

Interestingly, in the structure of the human glycine receptor α3 receptor of the Cys-loop receptor superfamily in complex with an allosteric modulator of AM-3607[33], AM-3607 binds to the subunit interface at the top of the central vestibule in the extracellular domain, which is adjacent to the canonical agonist binding site. These structural features are conceptually similar to the allosteric site in this study, whereas P2X receptors and Cys-loop receptors are evolutionarily and structurally distant. On the other hand, in the case of the P2X4 allosteric site, depending on the size of the compounds (e.g., BAY-1797), each compound molecule can bind to each other at the center of the vestibule, whereas in the case of the glycine receptor this appears to be impossible because the vestibule is much wider than that of P2X receptors.

Taken together, our work provides the structural basis for the binding and modulation of the P2X4 receptor by its allosteric modulators, as well as mechanistic insights into their subtype specificity. These insights could contribute to the rational design and optimization of allosteric modulators for P2X4 receptors, whose physiological functions are associated with various diseases, particularly neuropathic pain.

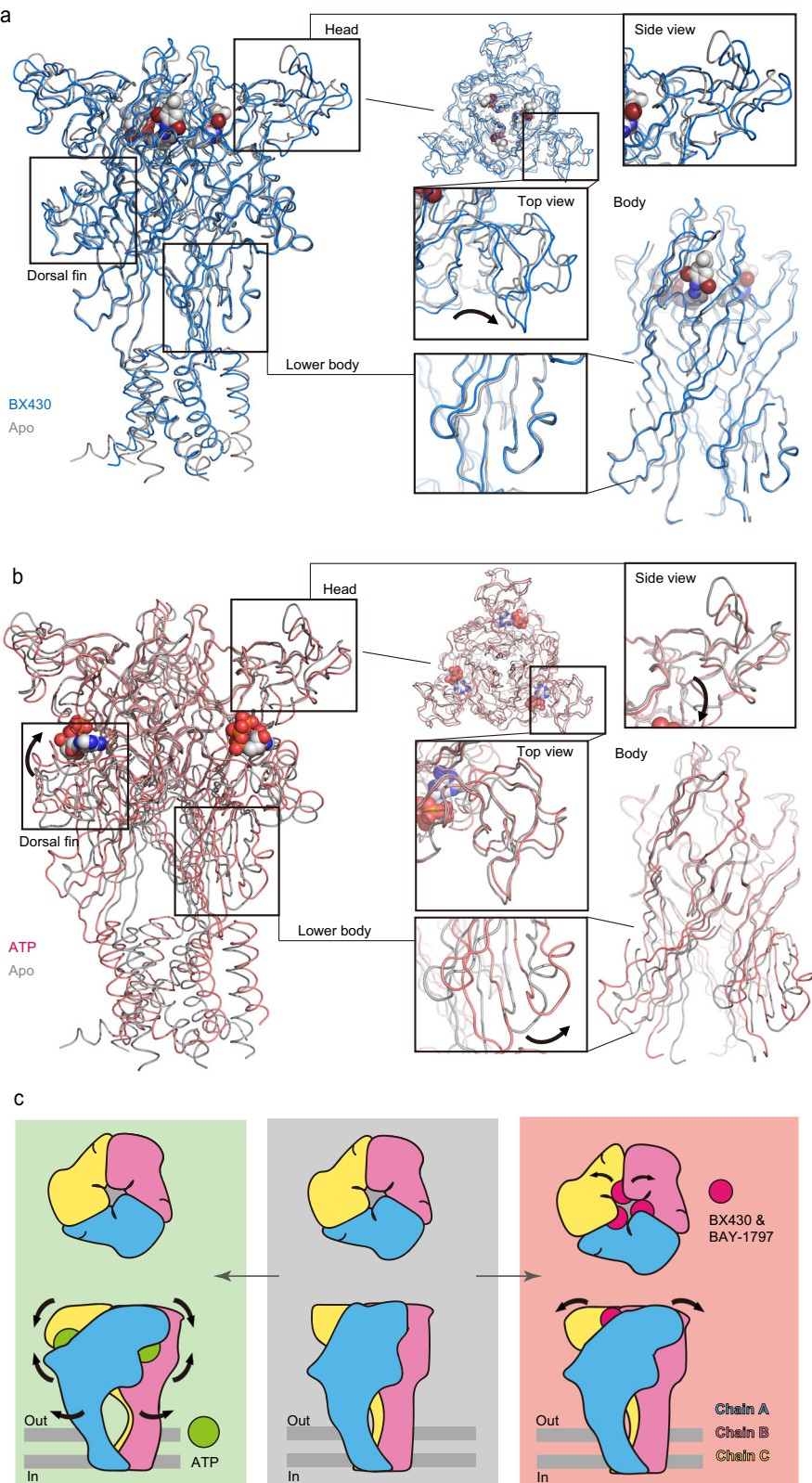

**Fig. 7 | Mechanism of allosteric inhibition. a, b** Superimposition of the BX430-bound (**a**) and ATP-bound (**b**) (PDB ID: 4DW1, red) zfP2X4 structures onto the apo structure. Close-up views of the head and body domains are also shown. Arrows indicate conformational changes. **c** Cartoon diagrams of the conformational changes associated with the binding of ATP (left) and allosteric modulators (right). The shape of each subunit follows a dolphin model, as shown in Supplementary Fig. 8. Arrows indicate structural changes between two structures.

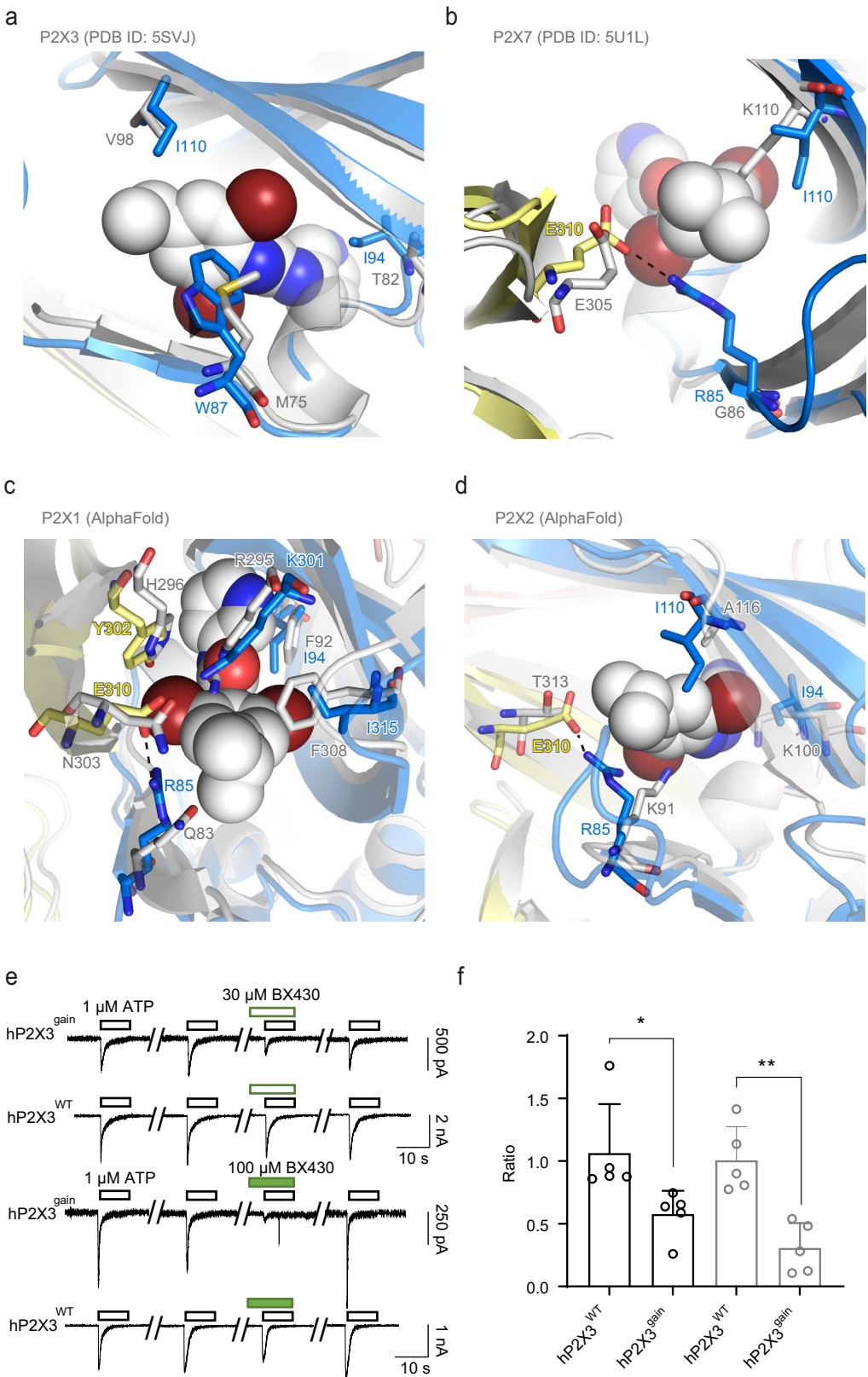

**Fig. 8 | Subtype specificity. a–d** Close-up view of the allosteric site of the BX430-bound zfP2X4 structure. The human P2X3 structure (PDB ID: 5SVJ) (**a**), panda P2X7 structure (PDB ID: 5U1L) (**b**), predicted structure of human P2X1 (AlphaFold) (**c**), and predicted structure of human P2X2 (AlphaFold) (**d**) are superposed and shown in gray. **e** Representative current traces of BX430 effects on ATP-evoked currents of human P2X3 wild type (WT) and its gain-of-function mutant (gain).

**f** Effects of 30 μM (black) and 100 μM (gray) BX430 on ATP (1 μM)-evoked currents of human P2X3 and its mutant (mean ± SD, $n = 5$). The $p$-value is 0.0361 and 0.0016, respectively (from left to right) (Two-sided unpaired $t$ test, $*p < 0.05$, $**p < 0.01$ vs. WT). The inhibition ratio is defined as in Fig. 4. $n$ is the number of biologically independent cells.

## Methods

### Expression and purification

In a previous electrophysiological analysis, the zebrafish P2X4 receptor was shown to exhibit ATP-dependent channel activity even with truncations of 27 residues at the N-terminus and 24 residues at the C-terminus[21]. In this project, we employed an expression construct of zebrafish P2X4 with the less drastic truncation of the N-terminus 8 residues and the C-terminus 7 residues (residues 9-382). The coding region (residues 9-382) of zebrafish P2X4 (GenBank: AAH66495.1) was synthesized (Genewiz, Inc., China) and subcloned into the pFastBac vector with a C-terminal human rhinovirus 3C (HRV 3C) protease cleavage site, coral thermostable GFP (TGP)[34,35] and Twin-Strep-tag. DH10Bac *Escherichia coli* cells (Gibco, USA) were used as the host for bacmid recombination. Sf9 cells were cultured in suspension at 27 °C in SIM SF culture medium (Sino Biological, China) and routinely passaged every other day. The initial recombinant baculovirus was generated by transfecting adherent Sf9 cells with the bacmid DNA using the FuGENE HD reagent (Promega, USA) and used to infect suspension cells for virus amplification. The protein was overexpressed in the Sf9 suspension culture at 27 °C for 60 h after virus infection. Cells were collected and lysed by sonication in TBS buffer (50 mM Tris-HCl, pH 8.0, 150 mM NaCl) with 1 mM phenylmethylsulfonyl fluoride (PMSF), 5.2 µg/mL aprotinin, 2 µg/mL leupeptin and 1.4 µg/mL pepstatin A. The supernatant was collected after centrifugation at $8000 \times g$ for 20 min and then ultracentrifuged at $200,000 \times g$ for 1 h. The membrane fraction pellets were solubilized in solubilization buffer [50 mM Tris-HCl, pH 8.0, 150 mM NaCl, 2% n-dodecyl-beta-d-maltopyranoside (DDM) (Anatrace, USA)] with 1 mM PMSF, 5.2 µg/mL aprotinin, 2 µg/mL leupeptin, 1.4 µg/mL pepstatin A and 0.2 unit/mL apyrase (Sigma, USA), stirred for 2 h. The solubilized mixture was centrifuged at $200,000 \times g$ for 1 h. The supernatant was loaded onto Strep-Tactin Superflow Plus beads (Qiagen, USA) pre-equilibrated with the wash buffer (100 mM Tris-HCl, pH 8.0, 150 mM NaCl and 0.05% DDM) and stirred for 1 h. The resin was washed with 10 CV of the wash buffer. The proteins were eluted by elution buffer (100 mM Tris-HCl, pH 8.0, 150 mM NaCl, 0.05% DDM, 2.5 mM desthiobiotin) and cleaved by HRV 3C protease to remove TGP and Twin-Strep tag in dialysis buffer (150 mM NaCl, 20 mM HEPES, pH 7.5, 0.05% DDM) overnight. The cleaved protein was applied to a Superdex 200 Increase 10/300 GL column (GE Healthcare, USA) pre-equilibrated with gel filtration buffer (150 mM NaCl, 20 mM HEPES, pH 7.5, 0.05% DDM). Peak fractions were collected and concentrated to ~2 mg/mL using an Amicon Ultra 50 kDa cutoff (Merck Millipore, USA). All purification steps were performed at 4 °C.

### Amphipol reconstitution

Purified P2X4 was mixed with PMAL-C8 amphipol (Anatrace, USA) dissolved in reconstitution buffer (150 mM NaCl, 20 mM HEPES, pH 7.5) at a ratio of 1:10 (w:w) and incubated overnight. Then, detergent was removed after 4 h of incubation with Bio-Beads SM-2 (Bio-Rad, USA) pre-equilibrated with the reconstitution buffer. The sample was applied to a Superdex 200 10/300 GL column (GE Healthcare, USA) pre-equilibrated with the reconstitution buffer. Amphipol-reconstituted P2X4 was pooled and concentrated to ~1 mg/mL using an Amicon Ultra 50 kDa cutoff (Merck Millipore, USA). All steps were performed at 4 °C.

### EM data acquisition

For grid preparation, amphipol-reconstituted P2X4 was incubated with 50 µM BX430 or 125 µM BAY-1797 at a final concentration for 1 h. Stock solutions of BX430 and BAY-1797 were prepared in DMSO at concentrations of 5 mM and 2.5 mM, respectively. No precipitation was observed after the addition of each compound, indicating that the compounds were completely dissolved.

After centrifugation at $100,000 \times g$ for 20 min, the sample was applied to holey carbon-film grids (Quantifoil, Germany, Au R1.2/1.3 µm size/hole space, 300 mesh) glow-discharged for 60 s, blotted with a Vitrobot (ThermoFisher Scientific, USA) with 100% humidity at 4 °C and plunge-frozen in liquid ethane cooled by liquid nitrogen.

Grids were transferred into a Titan Krios (Thermo Fisher Scientific, USA) electron microscope equipped with a K3 direct electron detector (Gatan Inc., USA). Cryo-EM images were collected at an acceleration voltage of 300 kV with a magnification of 29,000× for each dataset. For the dataset of the BX430-bound structure, the pixel size was 0.83 Å, and the defocus was varied from −1.3 µm to −2.0 µm. The dose rate was 20 e⁻/s, and the exposure time was 1.76 s, with an exposure of 50 e⁻/Å². For the dataset of the BAY-1797-bound structure, the pixel size was 0.89 Å, and the defocus was varied from −1.4 µm to −2.0 µm, and the dose rate was 18.5 e⁻/s. The exposure time was 1.83 s, with an exposure of 43 e⁻/Å². Detailed data collection statistics are shown in Table 1.

### Cryo-EM data processing

A dataset of 3866 movie stacks of P2X4 with BX430 and a dataset of 4929 movie stacks of P2X4 with BAY-1797 were collected. Motion correction, contrast transfer function (CTF) parameter estimation, particle picking, and further image processing were performed using RELION (version 4.0 for the dataset of the BX430-bound P2X4 receptor[36] and version 3.1 for the dataset of the BAY-1797-bound receptor[37]). Particles were extracted with a circle whose diameter was 120 Å and subjected to 2D classification using RELION. The extracted particle projections were imported into CryoSPARC[38] and subjected to ab initio reconstruction with C1 symmetry imposed. The well-defined subsets were selected and underwent nonuniform refinement and local refinement with C3 symmetry imposed.

For the dataset of P2X4 in complex with BAY-1797, additional processing was performed through 3D classification using a mask around the compound binding pockets. The best class was subjected to further nonuniform refinement and local refinement. After refinement and postprocessing, a final map for P2X4 in complex with BX430 at 3.23 Å resolution (using the Fourier shell correlation = 0.143 cutoff criterion) from 372,792 particles was obtained. The resolution of the final map for P2X4 in complex with BAY-1797 was 3.43 Å from 205,814 particles. Local resolutions were estimated using CryoSPARC[38]. The workflows for image processing and 3D reconstruction are shown. Figures were created using UCSF Chimera[39]. Notably, the cryo-EM data processed with C1 symmetry also showed the symmetric trimer of zfP2X4, but at lower resolutions (Supplementary Fig. 6).

### Modeling, refinement, and analysis

Models were built in Coot[40] using the zebrafish P2X4 receptor structure in the apo state as a template (PDB ID: 4DW0)[21]. Docking of the template model into the cryo-EM maps was performed in PHENIX[41]. The models were manually adjusted in Coot, followed by real-space refinement in PHENIX. Structure figures were generated using PyMOL (https://pymol.org/). The sequence alignment figure was generated using Clustal Omega[42] and ESPript 3.0[43]. The predicted structural models of human P2X1 (Supplementary Data 1) and P2X2 (Supplementary Data 2) were generated using AlphaFold[44] and ColabFold (v1.5.2)[45]. In ColabFold, the AlphaFold2 multimer (v3) was used with templates to generate the trimeric structures. After prediction, the predicted structure with the highest score was relaxed using amber force fields[46]. For both predictions of hP2X1 and hP2X2 trimers, the pLDDT and PAE values showed high confidence, especially in the core region of the P2X receptors (transmembrane domain and extracellular domain including the allosteric site in this study) (Supplementary Fig. 12). The structural model of human P2X4 was constructed based on the cryo-EM structure of zebrafish P2X4 in complex with BX430 using the program MODELLER (v10.4)[47].

## Table 1 | Cryo-EM data collection, refinement and validation statistics

| | BX430-bound zfP2X4 (EMD-36668) (PDB 8JV5) | BAY-1797-bound zfP2X4 (EMD-36669) (PDB 8JV6) |
|---|---|---|
| **Data collection and processing** | | |
| Magnification | ×29,000 | ×29,000 |
| Voltage (kV) | 300 | 300 |
| Electron exposure ($e^-/Å^2$) | 50 | 43 |
| Defocus range (μm) | −1.3 ~ −2.0 | −1.4 ~ −2.0 |
| Pixel size (Å) | 0.83 | 0.89 |
| Symmetry imposed | C3 | C3 |
| Initial particle images (no.) | 2,151,822 | 2,000,031 |
| Final particle images (no.) | 372,792 | 205,814 |
| Map resolution (Å) | 3.23 | 3.43 |
| FSC threshold | 0.143 | 0.143 |
| Map resolution range (Å) | 1.9 ~ 37.4 | 2.0 ~ 7.79 |
| **Refinement** | | |
| Initial model used (PDB code) | | |
| Model resolution (Å) | 3.23 | 3.43 |
| FSC threshold | 0.143 | 0.143 |
| Model resolution range (Å) | 1.9 ~ 37.4 | 2.0 ~ 7.79 |
| Map sharpening $B$ factor ($Å^2$) | −190.1 | −199.6 |
| Model composition | | |
| Non-hydrogen atoms | 7480 | 7458 |
| Protein residues | 949 | 948 |
| Ligands | NAG:9, BX4(BX430):3 | NAG:12, BAY(BAY-1797):3 |
| $B$ factors ($Å^2$) | | |
| Protein | 77.52 | 79.91 |
| Ligand | 102.09 | 92.59 |
| R.m.s. deviations | | |
| Bond lengths (Å) | 0.003 | 0.003 |
| Bond angles (°) | 0.565 | 0.563 |
| Validation | | |
| MolProbity score | 1.43 | 1.51 |
| Clashscore | 5.16 | 7.00 |
| Poor rotamers (%) | 0.76 | 0.00 |
| Ramachandran plot | | |
| Favored (%) | 97.14 | 97.35 |
| Allowed (%) | 2.86 | 2.65 |
| Disallowed (%) | 0.0 | 0.0 |

## Electrophysiology

HEK293 cells were cultured in Dulbecco's Modified Eagle Medium (DMEM) (Gibco, USA) supplemented with 10% fetal bovine serum (FBS) (PAN-Biotech, Germany), 1% penicillin–streptomycin (Gibco, USA), and 1% Glutamax (Gibco, USA) at 37 °C with 5% $CO_2$. Plasmids were transfected into HEK293 cells by calcium phosphate transfection. Nystatin-perforated patch-clamp recordings were performed 24-48 h after transfection at room temperature $(25 \pm 2 °C)$[48–50]. Patch pipettes were pulled from glass capillaries by a two-stage puller (PC-100, Narishige, Japan), and the resistance was 3−5 megaohms (mΩ) when filled with the pipette solution containing (in mM) 75 $K_2SO_4$, 55 KCl, 5 $MgSO_4$ and 10 HEPES adjusted to pH 7.2. The bath solution contained (in mM) 150 NaCl, 0.5 KCl, 10 glucose, 2 $CaCl_2$, 10 HEPES and 1 $MgCl_2$ adjusted to pH 7.35−7.40. Current traces were filtered at 2 kHz and acquired at 10 kHz via a Digidata 1550B interface and pCLAMP software (Molecular Devices, USA). The membrane potential was held at −60 mV during the recording. ATP, BX430 or BAY-1797 was dissolved in bath solution and applied to the Y-tube. Between each application, the cells were perfused for 6−8 min to allow for full recovery from desensitization.

## Molecular dynamics simulation

All atom molecular dynamics simulations were carried out by the program DESMOND[51,52]. The disordered parts of the TM helices of P2X4 in the cryo-EM structures were manually extended based on the apo state structure of P2X4 (PDB ID: 4DW0). All P2X4 structures with or without ligands were embedded in a palmitoyl-oleoyl-phosphatidylcholine (POPC) lipid bilayer and dissolved in simple point charge (SPC) water molecules. Periodic boundary conditions were applied with a buffer distance of 10 Å to each dimension of the simulation box. The systems were neutralized by adding counter ions ($Na^+$ or $Cl^-$) to balance the net charges. NaCl (150 mM) was added into the simulation system to represent background salt under physiological conditions. The DESMOND default relaxation protocol was applied to each system prior to the simulation run: (1) 100 ps simulations in the NVT ensemble with Brownian kinetics using a temperature of 10 K with solute heavy atoms constrained; (2) 12 ps simulations in the NVT ensemble using a Berendsen thermostat with a temperature of 10 K and small-time steps with solute heavy atoms constrained; (3) 12 ps simulations in the NPT ensemble using a Berendsen thermostat and barostat for 12 ps simulations at 10 K and 1 atm, with solute heavy atoms constrained; (4) 12 ps simulations in the NPT ensemble using a Berendsen thermostat and barostat at 300 K and 1 atm with solute heavy atoms constrained; and (5) 24 ps simulations in the NPT ensemble using a Berendsen thermostat and barostat at 300 K and 1 atm without constraint. After equilibration, the MD simulations were performed for 1.0 μs. For the repeats, two additional MD runs using different random seeds were performed for 0.5 μs (Supplementary Figs. 13, 14, 15). Long-range electrostatic interactions were computed using a smooth particle mesh Ewald method. The trajectory recording interval was set to 200 ps, and other default parameters of DESMOND were used during the MD simulation runs. The OPLS-2005 force field was used to model all atoms and their interactions[53,54]. All simulations were performed on a DELL T7920 graphic workstation with an NVIDA Tesla K40C-GPU. Preparation, analysis, and visualization were performed on a 12-CPU CORE DELL T3610 graphic workstation.

## Statistics and reproducibility

Electrophysiological recordings were repeated 3-9 times. Error bars represent the standard error of the mean. Cryo-EM data collection and refinement statistics are summarized in Table 1.

## Reporting summary

Further information on research design is available in the Nature Portfolio Reporting Summary linked to this article.

## Data availability

The atomic coordinates of the P2X4 structures were deposited in the Protein Data Bank under accession codes 8JV5 (BX430) and 8JV6 (BAY-1797). Cryo-EM maps were deposited in the Electron Microscopy Data Bank (EMDB) under accession codes EMD-36668 (BX430) and EMD-36669 (BAY-1797). All other relevant data are included in the paper or its supplementary information files, including the source data file (Source Data), or deposited in ScienceDB (https://doi.org/10.57760/sciencedb.09897) (https://doi.org/10.57760/sciencedb.11154). Source data are provided with this paper.

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

## Acknowledgements

We thank the staff scientists at the Center for Biological Imaging, Institute of Biophysics, and National Center for Protein Science Shanghai (Chinese Academy of Sciences) for technical assistance with cryo-EM data collection (project numbers: CBIapp202007004; 2020-NFPS-PT-005280). We also appreciate the kind support of Prof. Ye Yu (China Pharmaceutical University) as a senior mentor to Jin Wang. This work was supported by funding from STI2030-Major Projects (No. 2022ZD0207800) to J.W. and the National Natural Science Foundation of China to M.H. (32071234, 32271244 and 32250610205) and J.W. (32000869). This work was also supported by the Innovative Research Team of High-level Local Universities in Shanghai, a key laboratory program of the Education Commission of Shanghai Municipality (ZDSYS14005), the Open Research Fund of State Key Laboratory of Genetic Engineering, Fudan University (No. SKLGE-2105), and JST, PRESTO of Japan to M.I. (JPMJPR20E1).

## Author contributions

C.S. expressed and purified P2X4 and performed cryo-EM experiments with assistance from Y.M.Z., D.S., X.T., and M.I. S.C., and M.H. performed model building. Y.Q.Z., W.C., and M.S. performed the electrophysiology experiments. J.W. performed the MD simulation. C.S., J.W., and M.H. wrote the manuscript. J.W. and M.H. supervised the research. All authors discussed the manuscript.

## Competing interests

The authors declare no competing interests.
