## [Peer Review File · Nature Communications]

REVIEWER COMMENTS

Reviewer #1 (Remarks to the Author):

The manuscript describes the first set of structures for zebrafish P2X4 in amphipol with allosteric inhibitors BX430 and BAY-1797 bound, which were known to have preferential effects on P2X4 with respect to other P2XRs but where few direct mechanistic insights have been elucidated. The authors show that both the modulators bind in the same pocket and share many interactions with the channel. However, specific interactions of BAY-1797 and main chain carbonyl groups of P2X4 in addition to hydrophobic interactions between the modulator molecules interacting with the 3-fold axis seems to make this molecule highly efficacious. The authors have also done MD simulations in absence and presence of modulators to show the stability of the binding interactions. Moreover, they use electrophysiology to show how the binding of inhibitors affects the P2X4 channel activity and the mutations in binding pocket of the protein affect the modulatory action of BX430. One of the more interesting features of these ligand interactions with the channel is the binding site needs to expand compared to the apo channel for the ligands to bind. Importantly, previous computational predictions for where BX430 and Bay-1797 bind appear to be completely invalid. Overall, this is a nice study that lays the foundation in P2X4 receptors for modifying drugs specific to this subtype and hopefully can be used in alleviating various disorders associated with this receptor. The following are issues/concerns the authors should address in revision.

1) How confident are the authors that the differences in the binding pockets for the two compounds when compared to Apo are the result of differences between unbound and bound complexes considering that the previous Apo structure is an X-ray structure in detergent and the present structures use amphipols and cryo-EM? Have the authors attempted to solve an Apo structure using the present conditions and approaches?

2) In Figure 1, the authors show the density of the modulator used to fit each ligand. At the resolution of 3.4-3.8 Å it would be recommended to show the density for the ligand along with that for residues in proximity of the modulators. Adding more panels in Figures 2 and 3 showing more extensive EM density around the ligand binding sites would help increase confidence in the models. The extent to which the structural data support the models generated is somewhat of a concern, given that the local resolution is not the highest where the ligand bind.

3) The dose-response relations provided in Supp Fig 1 for zf and h receptors employ 1 mM ATP in the former and 100 μM ATP in the latter. Was there a reason for this difference? If the antagonists were competitive (which they are not), this difference in agonist concentration would be expected to greatly

diminish the extent of inhibition by the antagonist. Similarly different ATP concentrations are used in Fig 4, which although consistent with Supp Fig 1, yields considerably diminished ratios for h vs zf. Also, some of the mutations the authors make truncate the side chain (to Ala) whereas others are considerably more drastic (to Trp or charge reversing). Could the authors provide a rationale for the changes they made and how it might impact their interpretations? Unless I missed it, the authors don't define 'ratio' in Fig. 4B,C and should. I presume it's the size of the ATP activated current in the presence of BX430 divided by that in its absence, but this should be explicitly defined. Finally, Fig 4 would hold considerably more weight if the authors could provide exemplar primary data for all of the positions studied.

4) Have the authors performed the dose response for BAY-1797 for zebrafish and human P2X4? It would be good to see the dose response curve in the supplementary figure 1 because it is the first time it has been used on zebrafish P2X4 and has been the basis for structural studies.

5) To support the claims of subtype specificities, have the authors tried making mutations corresponding to key residues found in any other P2XRs?

6) Based on the superimpositions of their structures the authors predict the lower affinity of these drugs in other P2XRs. Were the structures just superimposed or docking experiments were performed to observe the changes that could predict the binding of the modulators?

7) Could the authors discuss why both rat and mouse P2X4 are not sensitive to BX430?

8) The methods section would seem to need more information about how the models of P2X1 and P2X2 were generated using AlphaFold and ColabFold.

Reviewer #2 (Remarks to the Author):

Structural insights into the allosteric inhibition of P2X4 receptors

This manuscript describes two cryo-EM structures of the zebrafish P2X4 receptor in complex with (moderately potent) allosteric inhibitors (BX430 and BAY-1797) in a resolution of 3.4 and 3.8 Angstrom, respectively.

A common binding site in the extracellular domain (close to the binding site previously identified for allosteric P2X7 antagonists in a cryo-EM structure of the panda-P2X7 receptor) was identified.

The manuscript is well written. However, the data is not fully convincing at this point. It is particularly disappointing that the zfP2X4 receptor was used.

1. Why did the authors not work on the human P2X4 receptor? The zfP2X4 is activated by ATP only at very high (millimolar) concentrations, which may already be toxic for the cells, while the human receptor is much more sensitive.

2. It is surprising that the employed antagonists are not or only weakly active at the rat and mouse P2X4 receptors, but appear to be similarly potent at the zfP2X4 receptor as at the human P2X4 receptor. How can this be explained (the sequence comparison of the binding site does not provide an explanation).

3. The potency difference between BAY and BX is very small. The speculations as to why BAY is more potent, have to be removed because there is no real difference.

4. Expansion of the binding pocket: from 17 to 18.3/18.7 Angstrom: This is not a very big change. Please comment.

5. Fig. 4B and C: Explain the Y-axis ("ratio"). Were all mutants similarly responsive to ATP? What are the EC50 values for ATP at the wt and the mutants? Are they similarly activatable or are there differences between mutants? What about rat/mouse (which are supposed to be insensitive to BX but have the same amino acid residues in the proposed binding pocket)?

6. Can ATP bind to the P2X4 receptor at the same time as the allosteric antagonist? In radioligand binding studies, there was no competition between both ligands according to literature.

7. Since published mutagenesis data are not consistent with the obtained structures: Could it be possible that the antagonists have more than one binding site?

Minor points

- MD: All abbreviations have to be explained

- MD simulations: explain in the text how and how long it was performed. Was it repeated (triplicate)?

- How was BX and BAY dissolved for incubation with the receptor on the grid? Their solubility

is limited.

- Fig. 1: turn the structure of BAY to align with the modeled structure.

Reviewer #3 (Remarks to the Author):

The study by Shen, Cui and Zhang et al., investigates the allosteric inhibition of P2X4 receptors. Using a combination of electrophysiological assays, single particle cryo-EM structures and molecular dynamics simulations, the authors propose a model for selective inhibition of P2X4 receptors by two small molecule antagonists, BX430 and BAY-1797. The work presents a significantly important advance in our understanding of small molecule modulation within this family of cation channels. Importantly, the work extends our understanding of the allosteric modulation of ligand-gated ion channels, enabling interesting future comparisons with other important ion channel families. Overall, the work is well executed and clearly written and the results are described well. I am very supportive of publication.

Although the study is well executed, in my opinion, the major weakness is the lack of a clear mechanism for the allosteric effect of BX430 and BAY-1797 and a more concrete link between the properties of the drugs and the structural and functional data in the study.

For example, the authors cite several studies that show these drugs are specific for subtype 4 of this ion channel family. However, the mutational studies did not, to my understanding, identify where this selectivity originates. For example, in the discussion section, the authors focus on the comparison between zfP2X4 and panda P2X7 and identify a plausible role for the hydroxyl of Tyr295 in P2X7 in clashing with Lys301 in the drug binding pocket, thus explaining why the drugs do not antagonise P2X7. But why not test this in their electrophysiological assays? Surely this hypothesis could be tested with a Phe299Tyr mutation in zfP2X4?

Similarly, much of the mutational work, which is very nice, is fairly one-dimensional. i.e., it merely supports the structures (knock out X and reduce function Y). I did wonder whether, within the extensive set of mutations made, did any mutants show more interesting functions? For example, did any of the mutations have an impact on the IC50 curve for the drugs shown in SI Fig. 1A? Did any of the mutations impact normal gating behaviour? This last question is particularly interesting as it might suggest the allosteric pocket has other functions in P2X biology. Are all the endogenous ligands/regulators known? Or have these drugs targeted a cryptic regulatory binding site? In a similar track, do the drugs or mutations impact ATP binding?

Overall, I liked the study and support publication. However, I did feel the study fell a little short in connecting biology with the structures as well as it might. I didn't leave with the impression the study had identified the reason why BX430 and BAY-1797 are specific to subtype 4. Given this is the main aim of the study, maybe the authors can clarify this point in their revision and make it clear where the specificity originates.

It might be interesting within the discussion to compare the allosteric mechanism described here with those observed in other ligand-gated ion channels, at least superficially. Is the mechanism more or less like the ones described for other systems? Does the three-fold symmetry of this system have an important role to play in how this system is regulated by small molecules etc?

Minor comments:

If the system is allosteric, are you not concerned about imposing C3 symmetry in your map calculations? Is it possible the drugs distort the symmetry as their mode of action?

It would be helpful to include a schematic interaction map (similar to LigPlot) for the drug binding site in the supplementary or main figure, inc. bond distances.

Why do you include the spurious 2D classes in SI Fig. 2B?

Did the authors try Bayesian polishing to improve the resolution? Did the drugs change how the samples behaved in the ice compared to the WT protein? The overall workflow seems very simple, which is great, but I wondered if more could be wrung from the dataset. Did the authors try alignment-free classification with a mask around the drug-binding pocket for example?

REVIEWER COMMENTS

Reviewer #1 (Remarks to the Author):

“The manuscript describes the first set of structures for zebrafish P2X4 in amphipol with allosteric inhibitors BX430 and BAY-1797 bound, which were known to have preferential effects on P2X4 with respect to other P2XRs but where few direct mechanistic insights have been elucidated. The authors show that both the modulators bind in the same pocket and share many interactions with the channel. However, specific interactions of BAY-1797 and main chain carbonyl groups of P2X4 in addition to hydrophobic interactions between the modulator molecules interacting with the 3-fold axis seems to make this molecule highly efficacious. The authors have also done MD simulations in absence and presence of modulators to show the stability of the binding interactions. Moreover, they use electrophysiology to show how the binding of inhibitors affects the P2X4 channel activity and the mutations in binding pocket of the protein affect the modulatory action of BX430. One of the more interesting features of these ligand interactions with the channel is the binding site needs to expand compared to the apo channel for the ligands to bind. Importantly, previous computational predictions for where BX430 and Bay-1797 bind appear to be completely invalid. Overall, this is a nice study that lays the foundation in P2X4 receptors for modifying drugs specific to this subtype and hopefully can be used in alleviating various disorders associated with this receptor. The following are issues/concerns the authors should address in revision.”

We appreciate the positive response from Reviewer #1. We have addressed the specific comments below.

“1) How confident are the authors that the differences in the binding pockets for the two compounds when compared to Apo are the result of differences between unbound and bound complexes considering that the previous Apo structure is an X-ray structure in detergent and the present structures use amphipols and cryo-EM? Have the authors attempted to solve an Apo structure using the present conditions and approaches?”

We fully understand the concern raised by Reviewer #1. Following this comment, we performed MD simulation of the apo structure of zebrafish P2X4 embedded in lipids using the previously published crystal structure in the apo state (PDB ID: 4DW0) (**Figs. 6F-6I**) and added the associated description in the revised manuscript (From Page 12, Line 251 to Page 13, Line 265). The structure was stable during the MD simulation (**Fig. 6F**) and adapted the nonexpanded conformation of the binding pocket after the MD simulation (**Fig. 6G, H**). Consistently, we observed the expansion of the binding pocket and the associated movement of the head domain in both the experimental structures and the MD simulations (**Fig. 6H, 6I**). Therefore, as suggested by Reviewer #1, the environment in the structure determinations was different (detergent for the apo structure and amphipol for the BX430/BAY-1797-bound structures), and we did not perform cryo-

EM of zebrafish P2X4 in the apo condition. However, the MD simulations of the apo crystal structure and the BX430-bound cryo-EM structure were performed in lipid environments, and we still observed the similar difference in their conformation as in the experimental structures. Therefore, it is less likely that these structural differences come from the solution environment in structure determination (detergent and amphipol) or methods of structure determination (crystallization and cryo-EM).

“2) In Figure 1, the authors show the density of the modulator used to fit each ligand. At the resolution of 3.4-3.8 Å it would be recommended to show the density for the ligand along with that for residues in proximity of the modulators. Adding more panels in Figures 2 and 3 showing more extensive EM density around the ligand binding sites would help increase confidence in the models. The extent to which the structural data support the models generated is somewhat of a concern, given that the local resolution is not the highest where the ligand bind.”

As suggested in this comment, we generated new Figures (**Fig. 2B, 3B**). In addition, during the revision of our manuscript, based on the suggestion from Reviewer #3, we reprocessed our cryo-EM data and were able to improve the resolution of the BX430-bound structure from 3.4 Å to 3.2 Å and the BAY-1797-bound structure from 3.8 Å to 3.4 Å (**Supplementary Figs. 2 and 4 and Table 1**).

“3) The dose-response relations provided in Supp Fig 1 for zf and h receptors employ 1 mM ATP in the former and 100 uM ATP in the latter. Was there a reason for this difference? If the antagonists were competitive (which they are not), this difference in agonist concentration would be expected to greatly diminish the extent of inhibition by the antagonist. Similarly different ATP concentrations are used in Fig 4, which although consistent with Supp Fig 1, yields considerably diminished ratios for h vs zf. Also, some of the mutations the authors make truncate the side chain (to Ala) whereas others are considerably more drastic (to Trp or charge reversing). Could the authors provide a for the changes they made and how it might impact their interpretations? Unless I missed it, the authors don’t define ‘ratio’ in Fig. 4B,C and should. I presume it’s the size of the ATP activated current in the presence of BX430 divided by that in its absence, but this should be explicitly defined. Finally, Fig 4 would hold considerably more weight if the authors could provide exemplar primary data for all of the positions studied.”

Below are my answers to each question.

“The dose-response relations provided in Supp Fig 1 for zf and h receptors employ 1 mM ATP in the former and 100 uM ATP in the latter. Was there a reason for this difference? If the antagonists were competitive (which they are not), this difference in agonist concentration would be expected to greatly diminish the extent of inhibition by

the antagonist. Similarly different ATP concentrations are used in Fig 4, which although consistent with Supp Fig 1, yields considerably diminished ratios for h vs zf.”

We apologize for any misunderstanding caused by our poor explanation. The difference in the ATP concentrations tested for zebrafish P2X4 (1 mM) and human P2X4 (100 μ M) has a reasonable explanation. This is simply due to the difference in the known EC50 values of zebrafish P2X4 and human P2X4 for ATP. The EC50 of zebrafish P2X4 for ATP (~800 μ M, PMID: 19641588) is much higher than that of human P2X4 (~6 μ M, PMID: 10694247), so a concentration of 1 mM ATP was required to activate zebrafish P2X4. Finally, as Reviewer #1 pointed out, our antagonists are not competitive, so in principle, these differences should not affect the results of our electrophysiological recordings, especially since the BX430 concentrations tested for each WT and its corresponding mutants are consistent for comparison (zebrafish P2X4 WT and its mutants: 3 μ M) (human P2X4 WT and its mutants: 5 μ M).

“Also, some of the mutations the authors make truncate the side chain (to Ala) whereas others are considerably more drastic (to Trp or charge reversing). Could the authors provide a for the changes they made and how it might impact their interpretations?”

We appreciate the comment. As Reviewer #1 mentioned, we tested two types of mutations. For the mutation to the residues with smaller side chains (mostly to Ala), we

designed these mutants to reduce the side chain-mediated interactions between the receptor and BX430, which would weaken the binding affinity. For the other types of mutations, such as the introduction of a bulkier side chain (e.g., to Trp) or changing the charge of the side chains (e.g., from Glu to Lys), we designed these mutants to introduce repulsion between the receptor and BX430, which would also weaken the binding affinity. In addition to these two types of mutants, we tested five more mutants of human P2X4 to introduce residues found in other P2X subtypes (R82G, R82Q, I91T, E307N, E307T) in response to Reviewer #1's comment regarding subtype specificity (see below) (**Fig. 4D**). We have included these descriptions in the revised manuscript (Page 8, Lines 165-171).

“Unless I missed it, the authors don’t define ‘ratio’ in Fig. 4B,C and should. I presume it’s the size of the ATP activated current in the presence of BX430 divided by that in its absence, but this should be explicitly defined. Finally, Fig 4 would hold considerably more weight if the authors could provide exemplar primary data for all of the positions studied.”

Following this comment, in the revised manuscript, we have specified how we defined the ratio in the legend of **Fig. 4**. In addition, we have provided all numerical digital data needed to reproduce all graphs (**Supplementary Data 3**) together with the representative current traces for all WT and mutants and their original digital data (**Figs. 4 and 8**) (doi:10.57760/sciencedb.09897, Data private access link for reviewers:

<https://www.scidb.cn/s/IJRJR3>). To be noted, these would also satisfy the Journal Policy of *Nature Communications*.

“4) Have the authors performed the dose response for BAY-1797 for zebrafish and human P2X4? It would be good to see the dose response curve in the supplementary figure 1 because it is the first time it has been used on zebrafish P2X4 and has been the basis for structural studies.”

As suggested in this comment, we determined the dose response for BAY-1797 for zebrafish and human P2X4 receptors (**Supplementary Fig. 1**).

“5) To support the claims of subtype specificities, have the authors tried making mutations corresponding to key residues found in any other P2XRs?”

As suggested in this comment, we tested five more mutants of human P2X4 to introduce the types of residues in other P2X subtypes (R82G, R82Q, I91T, E307N, E307T) (**Fig. 4B, 4D**). In addition, we designed a gain-of-function mutant of human P2X3 to introduce BX430 sensitivity to human P2X3. Strikingly, the P2X4-like mutations of three residues of human P2X3 (M75W/T82I/V98I) conferred sensitivity to BX430 (**Fig. 8E and 8F**). We have added the corresponding description in the revised manuscript (Page 9, Lines 177-192) (From Page 15, Line 310 to Page 16, Line 345). In summary, we believe

that these further structure-based mutational analyses strengthened our work on subtype specificity and hope that these additional experiments will fulfill the expectations of Reviewer #1. If you have suggestions for additional mutants that need to be specifically tested, please let us know.

“6) Based on the superimpositions of their structures the authors predict the lower affinity of these drugs in other P2XRs. Were the structures just superimposed or docking experiments were performed to observe the changes that could predict the binding of the modulators?”

We appreciate the comment. We did not perform the docking simulation because other P2X subtypes do not have sensitivity to BX430, making such a docking simulation impractical. On the other hand, a simple superposition together with the sequence alignment can still be useful to design point mutations, as we did in this study.

“7) Could the authors discuss why both rat and mouse P2X4 are not sensitive to BX430?”

To address this comment, we added an explanation in the revised manuscript (Page 9, Lines 193-197).

“8) The methods section would seem to need more information about how the models of P2X1 and P2X2 were generated using AlphaFold and ColabFold.”

In response to this comment, we have added a description and a new figure to show the confidence of the AlphaFold models and provided the atomic coordinates and log files from ColabFold in the revised manuscript (**Supplementary Fig. 12 and Data 1, 2**) (Page 21, Lines 447-454).

Reviewer #2 (Remarks to the Author):

“This manuscript describes two cryo-EM structures of the zebrafish P2X4 receptor in complex with (moderately potent) allosteric inhibitors (BX430 and BAY-1797) in a resolution of 3.4 and 3.8 Angstrom, respectively. A common binding site in the extracellular domain (close to the binding site previously identified for allosteric P2X7 antagonists in a cryo-EM structure of the panda-P2X7 receptor) was identified.

The manuscript is well written. However, the data is not fully convincing at this point.

It is particularly disappointing that the zfP2X4 receptor was used.”

We appreciate the time and effort that Reviewer #2 spent on our manuscript. Please see our responses below.

“1. Why did the authors not work on the human P2X4 receptor? The zfP2X4 is

activated by ATP only at very high (millimolar) concentrations, which may already be toxic for the cells, while the human receptor is much more sensitive”

We appreciate the comment. Although we have performed structure-based mutation analysis of human P2X₄, we have not performed structural studies on the human P2X₄ receptor, as mentioned by Reviewer #2. This is because in previous studies (PMID: 16615909, PMID: 19641588), mammalian P2X₄ receptors, including the human P2X₄ receptor, were found to be unsuitable for structural analysis due to their low expression level and stability. Nevertheless, we believe that our current work is not limited to structural information on zebrafish P2X₄ but also provides valuable mechanistic insights into the allosteric modulation of the human P2X₄ receptor for the following reasons.

A. I recognize the difference in amino acid sequence and EC₅₀ value for ATP between zfP2X₄ and hP2X₄. Nevertheless, our cryo-EM structure of zfP2X₄ and sequence alignment with hP2X₄ showed that the amino acid residues in the allosteric site are highly conserved between zfP2X₄ and human P2X₄. In addition, to verify whether the structural information we obtained from zfP2X₄ is applicable to hP2X₄, we performed mutational analysis with hP2X₄ and showed that the allosteric site is indeed functionally conserved in hP2X₄ (**Fig. 4**).

B. Furthermore, during the revision of this manuscript, we performed MD simulation

of the hP2X4 receptor with BX430, which was modeled from our cryo-EM structure of zfP2X4 with BX430, and showed stable binding to BX430 at the allosteric site, as in the MD simulations of zfP2X4 structures (**Supplementary Figs. 9 and 10**) (Page 6, Lines 122-129). Thus, by combining the structure-based mutational analysis of hP2X4 and the MD simulation of hP2X4 based on the zfP2X4 structure, we successfully provided mechanistic insights into the allosteric modulation of hP2X4.

C. It is common to use zebrafish ion channels for structural studies. This is because in some cases, human proteins may be unsuitable for structural studies, as in the case of P2X4. There are many examples of zebrafish ion channel structures (TRPC4, PMID: 29717981) (TRPM2, PMID: 31431622) (CFTR, PMID: 27912062) (TREK1, PMID: 36841877) that likewise provide valuable structural insights into their mammalian orthologs.

In conclusion, we fully understand Reviewer #2's concern regarding our use of zebrafish P2X4 for structural studies, but despite the difference in species, we have successfully provided mechanistic insights into the human P2X4 receptor, not limited to zfP2X4.

“It is surprising that the employed antagonists are not or only weakly active at the rat

and mouse P2X4 receptors, but appear to be similarly potent at the zfp2X4 receptor as at the human P2X4 receptor. How can this be explained (the sequence comparison of the binding site does not provide an explanation).”

To address this comment, we added a description in the revised manuscript (Page 9, Lines 193-197).

“3. The potency difference between BAY and BX is very small. The speculations as to why BAY is more potent, have to be removed because there is no real difference.”

In accordance with this comment, we have deleted the corresponding text in the revised manuscript (Page 8, Line 159).

“4. Expansion of the binding pocket: from 17 to 18.3/18.7 Angstrom: This is not a very big change. Please comment.”

We appreciate the comment. As mentioned by Reviewer #2, the binding of compounds induced relatively small structural changes at the entrance region of the binding pocket. This would suggest that such small changes at the entrance of the binding pocket are sufficient to accommodate the compounds (**Fig. 6A**). We have added the corresponding descriptions in the revised manuscript (Page 11, Lines 236-238).

“5. Fig. 4B and C: Explain the Y-axis (“ratio”). Were all mutants similarly responsive to ATP? What are the EC50 values for ATP at the wt and the mutants? Are they similarly activatable or are there differences between mutants? What about rat/mouse (which are supposed to be insensitive to BX but have the same amino acid residues in the proposed binding pocket)”

“Fig. 4B and C: Explain the Y-axis (“ratio”).”

We have added an explanation of the definition of the Y-axis (ratio) in the legend of Figure 4.

“Were all mutants similarly responsive to ATP? What are the EC50 values for ATP at the wt and the mutants? Are they similarly activatable or are there differences between mutants?”

As pointed out by another reviewer (Reviewer #1), the apparent difference in EC50 values for agonists and agonist concentrations tested would not affect the inhibition assays in the case of noncompetitive inhibition, although it is crucial in the case of competitive inhibition. This is also a widely known fact (https://en.wikipedia.org/wiki/Competitive_inhibition,

https://en.wikipedia.org/wiki/Non-competitive_inhibition, e.g., PMID: 19196704).

In our study, both BX430 and BAY-1797 clearly showed noncompetitive binding in our cryo-EM structures (**Figs. 2 and 3 and Supplementary Fig. 8**). Consistently, BX430 was previously shown to be a noncompetitive inhibitor (PMID: 25597706, PMID: 36328074). In addition, the BX430 concentrations tested for each WT and its corresponding mutants are consistent for comparison (zebrafish P2X4 WT and its mutants: 3 μ M) (human P2X4 WT and its mutants: 5 μ M). Thus, to perform an inhibition assay with BX430 in our study to compare the effect of BX430 between WT and mutants, it would not be necessary to obtain EC50 values for the WT and all mutants. In fact, in the previously published mutational analysis of ion channels including P2X receptors with noncompetitive inhibitors, it was not necessary to obtain all EC50 values for both the WT and mutants (e.g., PMID: 29674445, PMID: 30803485, PMID: 31024257).

Nevertheless, we fully understand that it is important to show that all tested P2X mutants are at least still active as an ATP-gated ion channel in patch clamp recordings, so we have included representative current traces for the mutational analysis (**Figs. 4 and 8**).

“What about rat/mouse (which are supposed to be insensitive to BX but have the same amino acid residues in the proposed binding pocket)”

Since, as Reviewer #2 mentioned, rat and mouse P2X4 receptors are known to be

insensitive to BX430, we did not use rat and mouse P2X4 receptors for our patch clamp recording. The amino acid residues in the binding pocket are quite similar but not exactly the same, and Ile315 is implicated in the difference in sensitivity between species. We have included a discussion of Ile315 in the revised manuscript (Page 9, Lines 193-197).

“6. Can ATP bind to the P2X4 receptor at the same time as the allosteric antagonist? In radioligand binding studies, there was no competition between both ligands according to literature.”

We appreciate the comment. In our cryo-EM structures, upon the binding of BX430, there are no structural changes that may inhibit the binding of ATP. We included a new figure and associated description to show this in the revised manuscript (**Supplementary Fig. S7D**) (Page 6, Lines 120-121).

“7. Since published mutagenesis data are not consistent with the obtained structures: Could it be possible that the antagonists have more than one binding site?”

To the best of our knowledge, there are three published articles that include the mutational analysis of P2X4 with BX430. Among them, two papers characterized the importance of Ile315 in zfP2X4 (Ile312 in hP2X4) (PMID: 31024257, PMID: 36843952), which agrees well with our cryo-EM structures, and I have included the corresponding

description in the revised manuscript, as mentioned above (Page 9, Lines 193-197).

In another paper, the authors predicted the cavity at the dorsal fin region to be a potential binding site for BX430 and performed mutational analysis, which showed that the mutation of Ile209 to Ala decreased BX430 sensitivity (PMID: 36328074). However, in our cryo-EM map, we did not find the EM density for BX430 in the dorsal fin region but did find that the side chain of Ile212, corresponding to Ile209 in human P2X4, has extensive hydrophobic contacts with the neighboring residues (**Supplementary Fig. 11**). Therefore, a possible explanation for the reduced BX430 sensitivity of the I209A mutant could be that a mutation at Ile209, which mediates extensive hydrophobic interactions with neighboring residues, could affect the folding of the protein, which could in turn indirectly affect the affinity to BX430 at the distant allosteric site in this study. Nevertheless, we do not completely exclude the possibility that another binding site for BX430 may exist, depending on the experimental conditions. We have included the relevant explanation in the revised manuscript (From Page 9, Line 197 to Page 10, Line 207).

“Minor points

- MD: All abbreviations have to be explained”

The abbreviation for molecular dynamics is MD. We have defined this abbreviation (Page 4, Line 82) and the other abbreviation, cryo-EM, in the revised manuscript (Page

4, Line 78).

“- MD simulations: explain in the text how and how long it was performed. Was it repeated (triplicate)?”

We performed each run for 1 μ s, as shown in the figures (**Figs. 5, 6 and Supplementary Figs. 9, 10**). In addition, to address this comment from Reviewer #2, during the revision of our manuscript, we performed two more repeats of MD runs, and each run was performed for 0.5 μ s (**Supplementary Figs. 13, 14, 15**).

“- How was BX and BAY dissolved for incubation with the receptor on the grid? Their solubility is limited.”

We fully understand Reviewer #2's concern regarding the low solubility of BX430 and BAY-1797. According to the manufacturer's protocol, both compounds are soluble at 10 mM in DMSO (BX430: <https://www.medchemexpress.com/bx430.html>) (BAY-1797: <https://www.medchemexpress.com/bay-1797.html>). We prepared stock solutions of BX430 and BAY-1797 in DMSO at concentrations of 5 mM and 2.5 mM, respectively, and added them to the amphipol-reconstituted zFP2X4 protein at final concentrations of 50 μ M (BX430) and 125 μ M (BAY-1797), respectively. With this method, we did not observe any precipitation after the addition of each compound, indicating that the

compounds were completely dissolved, whereas when we tested higher concentrations of BX430 and BAY-1797, we observed some precipitation due to the low solubility of the compounds. We have added these descriptions in the revised manuscript (Page 19, Lines 405-407).

“- Fig. 1: turn the structure of BAY to align with the modeled structure.”

To follow this comment, we turned the structure of BAY to align with the modeled structure (**Fig. 1**).

Reviewer #3 (Remarks to the Author):

“The study by Shen, Cui and Zhang et al., investigates the allosteric inhibition of P2X4 receptors. Using a combination of electrophysiological assays, single particle cryo-EM structures and molecular dynamics simulations, the authors propose a model for selective inhibition of P2X4 receptors by two small molecule antagonists, BX430 and BAY-1797. The work presents a significantly important advance in our understanding of small molecule modulation within this family of cation channels. Importantly, the work extends our understanding of the allosteric modulation of ligand-gated ion channels, enabling interesting future comparisons with other important ion channel families. Overall, the work is well executed and clearly written and the results are described well. I am very supportive of publication.”

We appreciate the positive response and support of our work for publication from Reviewer #3. We have addressed the specific comments below.

“Although the study is well executed, in my opinion, the major weakness is the lack of a clear mechanism for the allosteric effect of BX430 and BAY-1797 and a more concrete link between the properties of the drugs and the structural and functional data in the study. For example, the authors cite several studies that show these drugs are specific for subtype 4 of this ion channel family. However, the mutational studies did not, to my understanding, identify where this selectivity originates.

For example, in the discussion section, the authors focus on the comparison between zfp2X4 and panda P2X7 and identify a plausible role for the hydroxyl of Tyr295 in P2X7 in clashing with Lys301 in the drug binding pocket, thus explaining why the drugs do not antagonise P2X7. But why not test this in their electrophysiological assays? Surely this hypothesis could be tested with a Phe299Tyr mutation in zfp2X4?

Similarly, much of the mutational work, which is very nice, is fairly one-dimensional. i.e., it merely supports the structures (knock out X and reduce function Y). I did wonder whether, within the extensive set of mutations made, did any mutants show more interesting functions? For example, did any of the mutations have an impact on the

IC50 curve for the drugs shown in SI Fig. 1A? Did any of the mutations impact normal gating behaviour? This last question is particularly interesting as it might suggest the allosteric pocket has other functions in P2X biology. Are all the endogenous ligands/regulators known? Or have these drugs targeted a cryptic regulatory binding site? In a similar track, do the drugs or mutations impact ATP binding?

Overall, I liked the study and support publication. However, I did feel the study fell a little short in connecting biology with the structures as well as it might. I didn't leave with the impression the study had identified the reason why BX430 and BAY-1797 are specific to subtype 4. Given this is the main aim of the study, maybe the authors can clarify this point in their revision and make it clear where the specificity originates.”

We appreciate Reviewer #3's positive comment in support of our work for publication. We see several suggestions and questions raised by Reviewer #3, mainly to encourage us to strengthen the significance of our work, especially the subtype specificity. In response, we performed two additional experiments. First, we tested five more mutants of human P2X4 to introduce the types of residues found in other P2X subtypes (R82G, R82Q, I91T, E307N, E307T) (**Fig. 4B, 4D, 8A-8D**). In addition, we designed a structure-based mutant of another P2X subtype to introduce BX430 sensitivity. Strikingly, the P2X4-like mutations of three residues of human P2X3 (M75W/T82I/V98I) conferred sensitivity to BX430 (**Fig. 8A, 8E, 8F**). We have added the corresponding explanation in

the revised manuscript (From Page 15, Line 310 to Page 16, Line 345). We believe that these additional experiments, particularly the structure-based design of the BX430-sensitive P2X3 mutant, significantly strengthened our work on subtype specificity. In addition, our responses to each question are provided below.

“But why not test this in their electrophysiological assays? Surely this hypothesis could be tested with a Phe299Tyr mutation in zfP2X4?”

We apologize for the misunderstanding caused by our poor explanation. Our purpose in describing Tyr295/Phe299 (**Old Supplementary Fig. 10**) was not to explain subtype specificity but only to explain the difference in the shape of the binding pocket in the apo state due to the different orientation of the conserved Lys residue (Lys297 in pdP2X7 and Lys301 in zfP2X4). We did not intend to claim that this residue (Tyr295 in pdP2X7 and Phe299 in zfP2X4) is responsible for subtype specificity. Therefore, to avoid confusion, we have simply removed this figure and associated description from the revised manuscript.

“I did wonder whether, within the extensive set of mutations made, did any mutants show more interesting functions? For example, did any of the mutations have an impact on the IC50 curve for the drugs shown in SI Fig. 1A? Did any of the mutations impact normal gating behaviour? This last question is particularly interesting as it

might suggest the allosteric pocket has other functions in P2X biology. Are all the endogenous ligands/regulators known? Or have these drugs targeted a cryptic regulatory binding site? In a similar track, do the drugs or mutations impact ATP binding?”

We appreciate the suggestions and questions by Reviewer #3 that encouraged us to seek a new interesting aspect of our work. As far as we have observed, in addition to the lower affinity of P2X4 mutants for BX430, we did not find any unusual gating behavior in the mutants, which showed typical ATP-dependent activation and slow desensitization similar to the behavior of wild type, and we showed representative current traces of all mutants in the revised manuscript (**Fig. 4**). Among the ligands/regulators of P2X receptors, to the best of our knowledge, ATP is the only endogenous agonist, whereas several regulatory factors have been identified, including Zn^{2+} , Mg^{2+} and phosphoinositides. The binding sites for Zn^{2+} and Mg^{2+} have been structurally characterized (PMID: 26804916) (PMID: 31232692) but are different from the allosteric site in this study. Among phosphoinositides, the C-terminal region appears to be involved in binding, which also differs from the allosteric site in this study (PMID: 19036987) (PMID: 16857707). Regarding the effect of compounds on ATP binding, the previously reported radioligand binding assay showed that there was no competition between ATP and BX430 binding (PMID: 36328074). Consistently, as far as we have checked our cryo-EM structures, there are no structural changes upon binding of BX430 that could inhibit

the binding of ATP. We have prepared a new figure to show this and have included the figure and associated description in the revised manuscript (**Supplementary Fig. S7D**). (Page 6, Lines 120-121).

Overall, we appreciate the suggestions and questions from Reviewer #3 to help us strengthen the significance of our work. Based on this feedback, we have performed additional experiments, particularly regarding subtype specificity, which we believe strengthen the significance of our work, as I mentioned earlier.

“It might be interesting within the discussion to compare the allosteric mechanism described here with those observed in other ligand-gated ion channels, at least superficially. Is the mechanism more or less like the ones described for other systems? Does the three-fold symmetry of this system have an important role to play in how this system is regulated by small molecules etc?”

We appreciate these suggestions. Following this comment, we have explored the previously published structures of ligand-gated ion channels, such as Cys-loop receptors or ionotropic glutamate receptors. Unfortunately, many of the compounds bind to the transmembrane domain or act in a competitive manner, and allosteric binding to the extracellular domain is relatively rare. However, in the structure of the human glycine receptor $\alpha 3$ receptor in complex with a novel allosteric modulator of AM-3607 (PMID:

27991902), AM-3607 binds to the subunit interface at the top of the central vestibule in the extracellular domain, which is adjacent to the canonical agonist binding site. These structural features are conceptually similar to the allosteric site in this study, whereas P2X receptors and Cys-loop receptors are structurally distant. Interestingly, in the case of the P2X4 allosteric site, depending on the size of the compounds (e.g., BAY-1797), each compound molecule can bind to each other at the center of the vestibule, whereas in the case of the glycine receptor, this appears to be impossible because the vestibule is much wider than that of P2X receptors. We have included this description in the revised manuscript (From Page 16, Line 346 to Page 17, Line 354).

“Minor comments:

If the system is allosteric, are you not concerned about imposing C3 symmetry in your map calculations? Is it possible the drugs distort the symmetry as their mode of action?”

To address this comment, we processed our cryo-EM data with C1 symmetry (**Supplementary Fig. 6**) and found that the structures still showed the symmetric trimer, but at lower resolutions. We have included the new figure and accompanying description in the revised manuscript (Page 20, Lines 438-440).

“It would be helpful to include a schematic interaction map (similar to LigPlot) for the drug binding site in the supplementary or main figure, inc. bond distances.”

To address this comment, we have included figures for the schematic interaction map for the drug binding site in the revised manuscript (**Fig. 2C, 3D**).

Why do you include the spurious 2D classes in SI Fig. 2B?

To address this comment, we have removed them in the revised manuscript (**Supplementary Fig. 2B, 4B**).

“Did the authors try Bayesian polishing to improve the resolution? Did the drugs change how the samples behaved in the ice compared to the WT protein? The overall workflow seems very simple, which is great, but I wondered if more could be wrung from the dataset. Did the authors try alignment-free classification with a mask around the drug-binding pocket for example?”

We appreciate the helpful suggestions from Reviewer #3 on how to improve the data processing. Based on this comment, we tried both Bayesian polishing and alignment-free classification with a mask around the drug binding pocket, as well as other approaches. Overall, we were able to improve the resolution of the BX430-bound structure from 3.4 Å to 3.2 Å and the BAY-1797-bound structure from 3.8 Å to 3.4 Å using CryoSPARC with a mask around the drug binding pocket (**Supplementary Figs. 2, 3, 4, 5 and Table**

1). We have updated the corresponding descriptions in the revised manuscript and updated the coordinates deposited in the Protein Data Bank.

REVIEWERS' COMMENTS

Reviewer #1 (Remarks to the Author):

I have gone through the response to reviewers and the revised manuscript and I think the authors have done a really nice job of revising the manuscript to address the comments of the reviewers. The additional new data and other revisions have really strengthened the work. I also greatly appreciated the attitude of the authors and it is clear they have tried their best to address the concerns raised. The study is now ready and appropriate for publication.

Reviewer #2 (Remarks to the Author):

The manuscript by Hattori and coworkers has been improved.

It is, however, difficult to review the revised version because the authors often did not exactly describe in the rebuttal letter how they dealt with the criticism, for example, they just refer to the manuscript (see p. x, line y-z) without describing in the rebuttal letter what exactly has been changed.

Reviewer #3 (Remarks to the Author):

The authors have undertaken considerable additional work to strengthen the conclusions of this study. The new data in Figs. 4 & 8 illustrates the predictive power of the biochemical insights gained from the zf P2X4 structure and functional work. Overall, this study considerably extends our understanding of allostery in this family of LGICs and provides a plausible mechanism underlying the allosteric behaviour observed in the presence of BAY-1797 and BX430. The updated processing pipeline has also extended the resolution of the models and ligands, strengthening the study's structural data.

Minor comments - all editorial

- The density maps in the Figures appear a little messy and might be hard to see in the final paper. The authors may consider changing the representation.

- I would add 'Å' to the distances in the figures.

- Figure 4 - The axes in panels C and D are not the same width.

- Figure 7C - I would add more labels to this. Subunits, membrane, extracellular/intracellular etc. What do your arrows represent?

REVIEWER COMMENTS

Reviewer #3 (Remarks to the Author):

Minor comments - all editorial

“- The density maps in the Figures appear a little messy and might be hard to see in the final paper. The authors may consider changing the representation.”

We agree that some of the figures for the density maps, particularly those in Figures 2b and 3b, may be difficult to see. In response to this comment, we have removed the density maps from Figures 2b and 3b and created a new Supplementary Figures 11A and 11B to show the same information but with an improved view. Accordingly, the old Supplementary Figures 11A and 11B have now become Supplementary Figures 11C and 11D, and we have made the corresponding changes in the figure legends and figure numbering in the main text.

“- I would add 'Å' to the distances in the figures.”

Based on this comment, I added "Å" to the distances in Figures 2, 3, and 7.

“- Figure 4 - The axes in panels C and D are not the same width.”

To follow this comment, we made the width of the axes equal in Figures 4c and 4d.

“- Figure 7C - I would add more labels to this. Subunits, membrane, extracellular/intracellular etc. What do your arrows represent?”

In response to this comment, we have added more labels to Figure 7C. The arrows in Figure 7C indicate structural changes between two structures. We have made this more explicit in the figure legend.